# Forking Paths in Neural Text Generation

**Eric Bigelow**[1,2,3 †], **Ari Holtzman**[4], **Hidenori Tanaka**[2,3 *], and **Tomer Ullman**[1,2 *]

[1]Harvard University, Department of Psychology
[2]Harvard University, Center for Brain Science
[3]NTT Research, Physics & Informatics Lab
[4]University of Chicago, Department of Computer Science

## Abstract

Estimating uncertainty in Large Language Models (LLMs) is important for properly evaluating LLMs, and ensuring safety for users. However, prior approaches to uncertainty estimation focus on the final answer in generated text, ignoring intermediate steps that might dramatically impact the outcome. We hypothesize that there exist key *forking tokens*, such that re-sampling the system at those specific tokens, but not others, leads to very different outcomes. To test this empirically, we develop a novel approach to representing uncertainty dynamics across individual tokens of text generation, and applying statistical models to test our hypothesis. Our approach is highly flexible: it can be applied to any dataset and any LLM, without fine tuning or accessing model weights. We use our method to analyze LLM responses on 7 different tasks across 4 domains, spanning a wide range of typical use cases. We find many examples of forking tokens, including surprising ones such as punctuation marks, suggesting that LLMs are often just a single token away from saying something very different.

## 1 Introduction

Large Language Models (LLMs) demonstrate impressive yet opaque capabilities that emerge during next-word prediction (Brown et al., 2020; Kaplan et al., 2020; Bubeck et al., 2023), and a good deal of current research is devoted to understanding and interpreting LLM behavior (Chang et al., 2024; Anwar et al., 2024; Bricken et al., 2023; Holtzman et al., 2023; Akyürek et al., 2022). LLMs are often treated as black boxes due to the sheer complexity of their internal workings, and because many state-of-the-art models are only accessible at the level of inputs and outputs. One way to assess any dynamic system is to consider what possible things it *could* have done, but didn't. In text generation, we can liken a text sequence to a path the system took through the semantic space of all possible paths, and ask: what other paths could the system have taken? Are there key points where re-sampling the system at that specific point, but not others, would lead to very different paths?

Work on uncertainty estimation in LLMs tackles the related problem of assessing how likely an LLM is to respond with different final answers, e.g. the probability of responding *"A"* or *"B"* to a multiple choice question (Kadavath et al., 2022; Tian et al., 2023; Guo et al., 2017; Ye et al., 2024). Previous approaches to black-box uncertainty estimation have yielded important insights by analyzing data such as the logit probabilities of the final tokens in an LLM's output, or the fraction of text responses that end in the correct answer (Geng et al., 2024; Xiong et al., 2024). However, in our analogy, these approaches consider only the final destination, and not the paths leading to them.

A major limitation of prior work on uncertainty estimation is that the last few tokens of an LLM's output are largely determined by previous tokens. For example, a single wrong step when solving a multi-step reasoning problem (e.g. *"The current year is 2021 . . . "*) can cascade into a wrong final answer (e.g. *". . . The current British head of state is Queen Elizabeth."*), or other undesired responses (Zhang et al., 2023). Uncertainty over intermediate tokens or reasoning steps will not be reflected in the final tokens of the LLM's response, since these tokens will be nearly deterministic (100%

---

[†] Correspondence to: `ebigelow@g.harvard.edu`
[*] Equal contribution

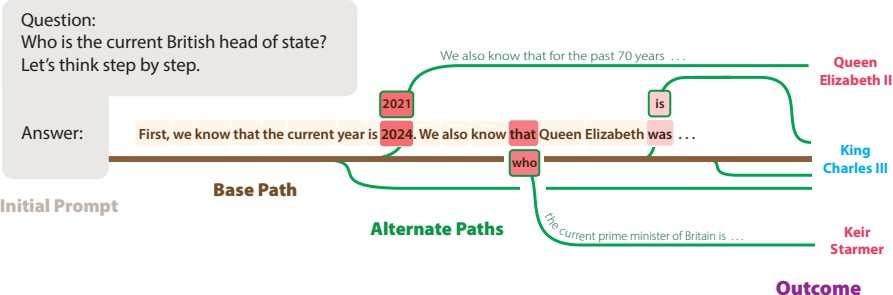

Figure 1: **Forking paths in text generation: can a single token alter the outcome?** At each step of next-word text generation, an LLM has some probability of sampling a variety of possible next tokens. This raises the question: are there specific *forking tokens*, where choosing a certain token over other probable alternatives results in a distinct outcome? To test this hypothesis, we estimate uncertainty in text generation by systematically re-sampling alternate completions at each token in a single *base path*, such as a greedily decoded sequence, to identify if there are forking tokens where two paths diverge into different outcomes. We find forking tokens where expected content words (*2021/2024* in this example) lead to a different final answers (*Queen Elizabeth* or *King Charles*), but we also find forking tokens in unexpected places (e.g. *that/who*).

confidence) given the rest of the text preceding them. A similar assumption is made in process-level supervision (Lightman et al., 2023), which gives an LLM feedback for the correctness of each step of its solutions, in addition to its final answer (i.e. outcome-level supervision). Perhaps, then, we might gleam valuable insights by analyzing uncertainty in paths and not just outcomes.

Our approach is to study *uncertainty dynamics*, or how an LLM's likelihood of producing different responses changes as each new token is generated (Fig. 1). Specifically, we propose the Forking Tokens Hypothesis: that in LLM text generation, there will be individual *forking tokens* which, if generated, lead to dramatic differences in subsequent text (Fig. 1). Uncertainty dynamics and forking tokens are unseen by prior approaches to 'static' uncertainty estimation such as taking the logits of the final answer token, or re-sampling many full responses. This inspires a new 'dynamical' way of thinking about uncertainty in text generation, where we study the influence that individual tokens have on the eventual outcome. We develop a methodology called Forking Paths Analysis (Sec. 2) in order to shed light on uncertainty dynamics and to empirically test for forking tokens. We find dramatic uncertainty dynamics in GPT-3.5 in many tasks commonly used for evaluation, including single tokens that cause the model to suddenly flip from low confidence to high confidence in a final answer. This supports the Forking Tokens Hypothesis, and suggests that uncertainty dynamics in GPT-3.5 are considerably more chaotic than high confidence final answers might suggest.

Put briefly, our primary contributions are:

1. **A novel hypothesis regarding the existence of 'forking tokens' that greatly impact the outcome of text generation**. We propose the Forking Tokens Hypothesis, that there exist individual tokens which, if sampled during decoding instead of an alternative token, lead to dramatic differences in subsequent text generation (Fig. 1).

2. **A novel approach to representing uncertainty at each token in next-word prediction**. Our method aggregates text samples into time series and conditional distributions, revealing uncertainty *dynamics* invisible to prior work (Sec. 2.1, 2.2). We use change point detection models and survival analysis to empirically test our hypothesis and efficiently scale across hundreds of individual analyses (Sec. 2.3, 2.4).

3. **Our analysis shows striking text generation dynamics, including change points in many sequences and unexpected forking tokens such as space characters** We examined text generation dynamics in GPT-3.5 using 7 different LLM evaluation tasks (Sec. 4). Our results support the Forking Tokens Hypothesis.

## 2 FORKING PATHS ANALYSIS

In text generation, exchanging a single token may drastically alter subsequent text. This is clearly true in reasoning. For example, if we ask *"Who is the current British head of state?"*, the intermediate

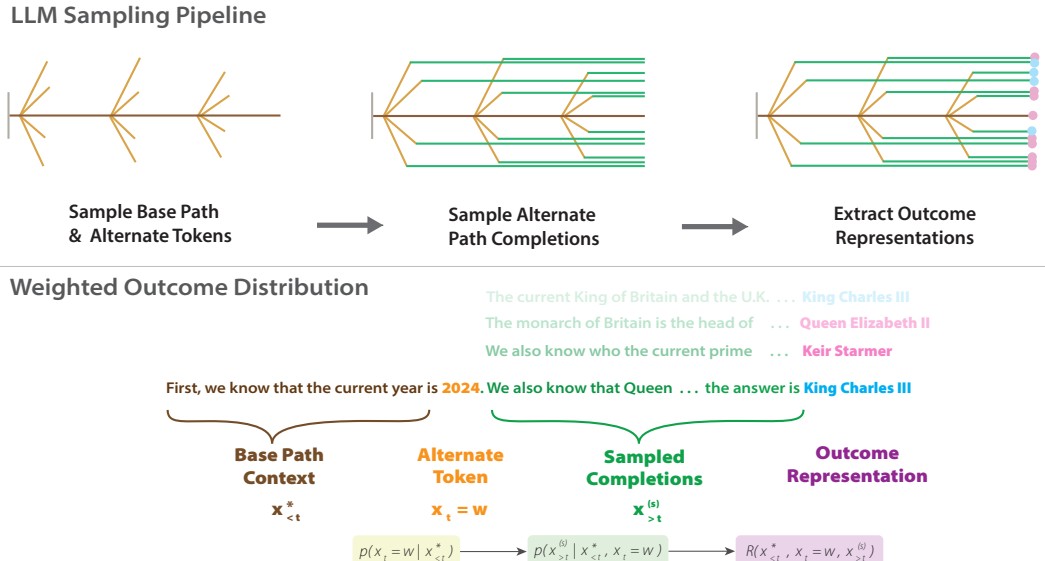

Figure 2: (Top) **Systematically sampling alternate paths in text generation.** Our data collection proceeds in three stages. First, we decode a base path text completion $x^*$ from an LLM given some prompt and record the most probable alternate tokens $w$ at each step $t$. Next, we re-sample $S$ completions $x_{>t}^{(s)}$ by appending $x_{<t}^*$ to the original prompt, along with each alternate token $x_t = w$. Finally, we extract outcome vector representations $R(\cdot)$ for each sample. In our experiments, for $R$ we use a different LLM to extract the final answer from a path. (Bottom) **Probability-weighted outcome distributions** We aggregate outcome representations $R(\cdot)$ into outcome distributions $o_t$ and $o_{t,w}$ by weighting each outcome with next-token probabilities for the forking token $p(x_t = w \mid x_{<t}^*)$ and for the sampled path completion $p(x_{>t}^{(s)} \mid x_{<t}^*, x_t = w)$. Also see Eq. 1.

reasoning step *"The current year is (2021/2024) . . . "* can lead to different final answers. This could also occur in open-ended text generation, when a single token distinguishes topics. For example *"Billy woke up in a hotel . . . "* and *"Billy woke up in a spaceship . . . "* leads to very different stories. LLMs generate text one token at a time, and sampling a word such as *"hotel"* instead of *"spaceship"* could steer autoregressive text generation towards one path over another. Many possible tokens could cause forking if exchanged, e.g. if we manually set the token in the previous example to *"coconut"*. However, we aim study the forking tokens that are most likely to be sampled during text generation.

We expect text generation to be relatively stable, in that most tokens an LLM is likely to sample would only change surface features and would not affect the outcome or overall meaning. However, LLMs also show significant biases in producing certain words and phrases over others (McCoy et al., 2023), and adapting their semantic priors on the fly (Wei et al., 2023). We also might expect, then, to find unexpected forking tokens with LLMs, for example if *"Billy woke up in a . . . "* leads to a story about a summer vacation and *"Billy woke up in the . . . "* leads to a science fiction story. This could occur if an LLM were considering two different outcomes to a story (e.g. a summer vacation or a sci-fi story), but these outcomes were biased towards specific words or phrases such as *"Billy woke"* and *"Billy awoke"*. If we find unexpected forking tokens such as this, it might suggest that LLMs are not planning their final responses before generating text, and instead they are effectively deciding what the outcome will be as each new token is sampled.

We predict that in text generation with LLMs, there will be individual tokens where text generation 'forks' into multiple distinct outputs. More formally, we propose the Forking Tokens Hypothesis: that a single token being exchanged for a probable alternate token can significantly impact the outcome of subsequent text generation in LLMs. To test this, we develop a method that includes a multi-stage sampling pipeline (Section 2.1), aggregating text samples into outcome distributions (2.2), and applying statistical models to test our hypothesis (2.3, 2.4).

## 2.1 LANGUAGE MODEL SAMPLING PIPELINE

For our analysis, we collect a large number of output texts from an LLM for each individual question prompt using a 3-stage sampling pipeline (Fig. 2, Top). We begin by decoding a single *base path*

response $x^*$ to a given prompt. This sample includes, for each token $x_t$ in sequence, the logit probabilities $p(x_t = w|x_{<t})$ for the top $K$ tokens $w$ at index $t$. Next, for each index $t$ and each alternate token $w$ with sufficiently high probability $p(x_t = w \mid x^*_{<t})$, we re-sample a batch of samples $\{x^{(1)}_{>t} \ldots x^{(S)}_{>t}\}$ by prompting the LLM with the original input text appended to the base path response $x^*_{<t}$ up to time $t$, with the last token being $x_t = w$. In the third step, we extract an outcome representation for each sample $R(x^*_{<t}, x_t = w, x^{(s)}_{>t})$. Described in the following section, $R$ is a semantic vector representation such as a one-hot encoding of the final answer.

These stages rely on token logit probabilities from the evaluated LLM, which are available in black-box LLM APIs such as OpenAI and TogetherAI and can also be inferred for APIs without this feature (Morris et al., 2023). In our experiments, the third stage $R$ uses a second LLM, which may be the same or different from the main LLM being evaluated, and does not require logit probabilities. We prompt this model to extract the final answer from $x^*$, and convert its response into a categorical value, such as *"A"* or *"B"*.

## 2.2 REPRESENTING TEXT GENERATION OUTCOMES

We construct **outcome distributions** for individual token indexes $o_t$ and for token values $o_{t,w}$. We define an outcome distribution $o$ as the expected value of a semantic vector representation $R$, which varies depending on the task and takes as input both the question prompt and model-generated response. In the case of a multiple-choice task, $R$ is a one-hot encoding of the final answer in the response string (e.g. *'A'*, *'B'*, *'C'*, or *'D'*). For open-ended tasks without final answers, $R$ can be any arbitrary semantic vector embedding.

$$
\begin{aligned}
o_{t,w}(x^*) &= \mathbb{E}_s\big[R(x^*_{<t},\ x_t = w,\ x^{(s)}_{>t})\big] \\
&= \sum_s p(x^{(s)}_{>t} \mid x^*_{<t},\ x_t = w)\ R(x^*_{<t},\ x_t = w,\ x^{(s)}_{>t})
\end{aligned}
\tag{1}
$$

$$
\begin{aligned}
o_t(x^*) &= \mathbb{E}_w\big[o_{t,w}(x^*)\big] \\
&= \sum_w p(x_t = w \mid x^*_{<t}) \sum_s p(x^{(s)}_{>t} \mid x^*_{<t}, x_t = w)\ R(x^*_{<t},\ x_t = w,\ x^{(s)}_{>t})
\end{aligned}
\tag{2}
$$

Outcome Distr.     Next-Word Prediction     Sample Probability     Outcome Representation

In our experiments, $o_t$ and $o_{t,w}$ are histograms over categorical outcome representations $R$, weighted by the sample probability $p(x_{>t}|x_{\le t}) = \prod_{t'=1}^{t} p(x_{t'}|x_{1:t'})$. In the case of $o_t$, outcome representations are also weighted by the forking token probability $p(x_t = w|x_{<t})$. All such weighting probabilities are derived from next-token prediction probabilities $p(x_t|x_{1:t-1})$. For brevity, we refer to $x_{1:t-1}$ as $x_{<t}$ and $x_{t+1:T}$ as $x_{>t}$. Our approach is Bayesian, in that we weight samples according to a graph of conditional probabilities and we interpret output certainty as the degree of an LLM's belief. This stands in contrast to frequentist calibration methods, which interpret certainty as the fraction of problems which are answered correctly (Guo et al., 2017; Kadavath et al., 2022; Geng et al., 2024), $o_t$ and $o_{t,w}$ have the advantages of being easy to visualize, and suitable for statistical analysis. $o_t$ can be represented as a multivariate time series (Figs. 4, 5), and $o_{t,w}$ can be plotted with parallel sets diagrams (Fig. 7). These visualizations reveal uncertainty dynamics across tokens, showing how the outcome distribution can dramatically change over the course of text generation.

Next, in the following sections we describe methods for analyzing outcome distributions $o_t$ and $o_{t,w}$ to test for forking tokens. We define a **forking token** $x_t$ as a token index $t$ or value $x_t = w$ which, if exchanged, leads to a very different outcome (or 'path'). In the case of $o_t$, a forking token index $t$ corresponds to an abrupt change in the time series $o_t$, i.e. where $o_{>t} \not\approx o_{<t}$. To identify these changes, we use Bayesian change point detection models, a class of statistical models that empirically test whether there are abrupt changes in a time series, and if so to identify the times $t$ when those changes occur. For $o_{t,w}$, a forking token value $x_t = w$ occurs when the outcome distribution for an alternate token $w$ deviates from the distribution for the base path token $w^*$ by at least $\epsilon$. To identify forking tokens in $o_{t,w}$, we use a discrete-time survival analysis where a hazard (i.e. non-survival) occurs when $[o_{t,w} \not\approx o_{t,w^*}]$ for some token $w$.

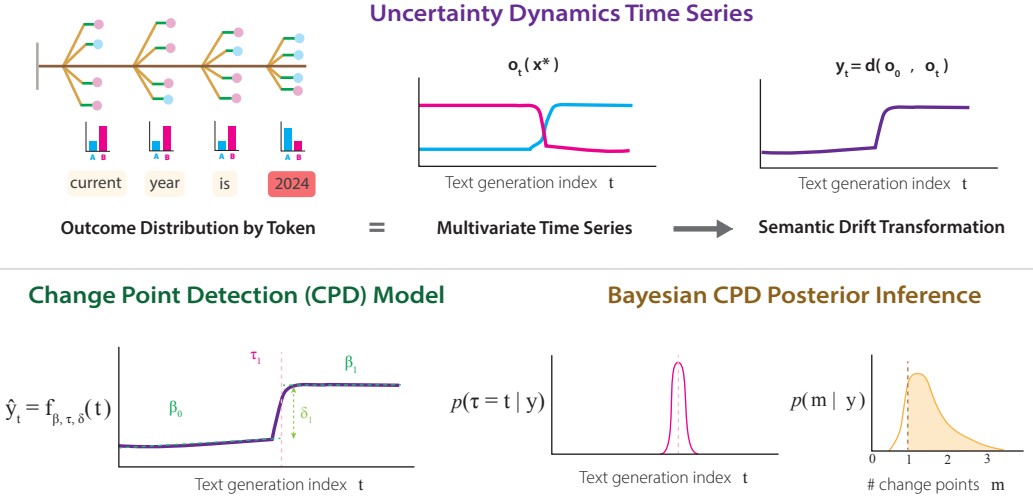

Figure 3: **Uncertainty dynamics time series** (Top) The outcome distribution $o_t$ is equivalent to a multivariate time series, where each possible outcome (e.g. *King Charles* or *Queen Elizabeth*) can be plotted as a separate line. To simplify our modeling, we convert $o_t$ into a univariate time series using a semantic drift transformation $y_t$. **Bayesian Change Point Detection model**. (Bottom, Left) We use a Bayesian Change Point Detection (CPD) model to identify sudden shifts in $y_t$. This model fits $y_t$ by splitting it into $m$ segments at times $\tau_i$, and fitting separate linear models with parameters $\beta_i, \delta_i$ to each segment $i$. (Bottom, Right) Our CPD model uses Monte Carlo sampling to infer posterior distributions that help us interpret analysis results: $p(\tau = t \mid y)$, or how likely it is that a change $\tau$ happened at time $t$, and $p(m \mid y)$, or how many change points, if any, occurred in the time series.

## 2.3 BAYESIAN CHANGE POINT DETECTION

We now present a method for automatically identifying tokens where $o_t$ changes suddenly, and for statistically testing the hypothesis that such tokens exist. In simple terms, our goal is to test for whether there are indexes $t$ such that the outcome distribution changes substantially before and after $t$, i.e. $o_{>t} \not\approx o_{<t}$. Change Point Detection (CPD) is an area of statistics that models time series data with abrupt changes to the line's intercept and/or slope. CPD models jointly infer what time $t$ a change occurs at, as well as trend parameters $\beta$ for time segments before and after the change. We suppose that there may be multiple forking tokens, or change points, in the outcome distribution $o_t(x^*)$ for a single sequence $x^*$. This leads us to use multiple CPD models, which assume there may be multiple change points. However, inference with multiple CPD models can be exponentially more expensive than with single CPD (Fearnhead, 2006). One solution to this complexity is using Bayesian models for efficient approximate inference. Since multiple CPD with multivariate time series (as in $o_t$) is a relatively young area of research (Cabrieto et al., 2017) with limited available tooling, we convert $o_t$ into a univariate time series $y_t$ using a *semantic drift* metric (Fig. 3, Top), as in Kulkarni et al. (2015). Each time point in the univariate time series $y_t = d(o_0, o_t)$ is the distance between the initial outcome distribution $o_0$ and outcome distributions for subsequent time steps $o_t$, given an arbitrary distance metric $d$. In our experiments, we use $L_2$ distance for $d$.

More specifically, a CPD model decomposes a time series $y$ into a set of $m$ segments. Each segment $i \in \{0 \dots m\}$ is fit by a regression model with intercept (i.e. abrupt change) $\delta_i$ and slope $\beta_i$, applied to time steps $t$ between $\tau_{i-1}$ and $\tau_i$. In a Bayesian CPD model, $p(\tau = t|y)$ describes how likely it is that a change point $\tau$ occurred at each time $t$ in time series data $y$, and $p(m|y)$ describes belief in the number of change points $m$ in a series. The beginning and end of a sequence are treated as fixed change points (i.e. $\tau_0 = 0$ and $\tau_{m+1} = T$) which are excluded from analysis of $p(\tau = t \mid m)$ and $p(m \mid y)$. We test for statistically significant change points by comparing the hypothesis that there is no change point ($m = 0$) with the hypothesis that there is at least one ($m \geq 1$) (Aminikhanghahi & Cook, 2017). In Bayesian CPD, this entails model comparison between a model with no change points and a model with at least one change point. We take a Bayes factor $p(m \geq 1|y) / p(m = 0|y)$ greater than 9 as supporting the hypothesis that there is at least 1 change point (Kass & Raftery, 1995) [*]. We use an extremely efficient open-source implementation of Bayesian CPD which utilizes

---

[*]9 is a typical threshold for Bayes factor significance testing, analogous to $\alpha = .05$ in frequentist statistics

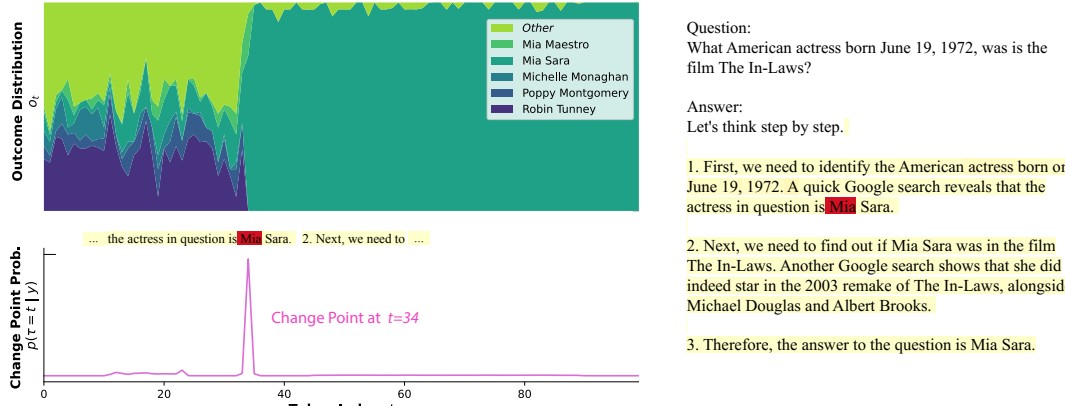

**Figure 4: The outcome distribution can collapse after a single step in chain-of-thought reasoning.** (Left, Top) The outcome distribution $o_t$ and (Left, Bottom) change point probabilities $p(\tau = t|y)$ for a single question from HotpotQA and a greedily decoded base path $x^*$. (Right) Tokens in $x^*$, highlighted according to $p(\tau = t|y)$, where yellow indicates low probability of a change point, red indicates high probability; the initial prompt is shown above $x^*$ with no coloring. We see striking uncertainty dynamics in $o_t$: the outcome distribution remains stable with the top single answer being *Robin Tunney* (the correct answer) until the forking token $t = 34$, when the distribution suddenly collapses to *Mia Sara* (a hallucinated wrong answer).

Monte Carlo Gibbs sampling to infer posterior distributions $p(m \mid y)$ and $p(\tau = t|y)$ given a sequence $y$ (Zhao, 2019). Further details of CPD model are provided in App. B.

## 2.4 SURVIVAL ANALYSIS

We perform a survival analysis to estimate on how likely it is that text generation would be greatly impacted if a token in a base path $x_t^* = w^*$ were instead sampled as $w$. This describes cases where an alternate token $w$ leads an LLM down a very different path from $x^*$, as in Fig 7 (Right). This forking may not appear as a stable change in $o_t(x^*)$ (Sec 2.3) since $w$ was not actually sampled and therefore does not impact $o_{>t}(x^*)$. We define the *survival function* $S(t)$ as the probability that the base path "survives" sampling alternate tokens $w$ that would change the outcome distribution:

$$S(t) = 1 - \prod_{t'=1}^{t} \mathbb{E}_w \left[ o_{t',w} \not\approx o_{t',w^*} \right] \tag{3}$$

$$= 1 - \prod_{t'=1}^{t} \sum_w p(x_{t'} = w | x_{<t'}^*) \, \mathbb{1}\left[ d(o_{t',w}, o_{t',w^*}) > \epsilon \right]$$

$S(t)$ is a discrete time survival function $S(t) = 1 - \prod_{t'=1}^{t} h(t')$ where $h(t)$ is the probability that a failure (or *hazard* $h$) occurs at time $t$. In our case, a failure is when an alternate token causes the outcome distribution to shift significantly from the greedy, i.e. $o_{t,w} \not\approx o_{t,w^*}$, which we estimate by testing whether the distance between outcome distributions $d(o_{t,w}, o_{t,w^*})$ is greater than some threshold $\epsilon$. $d$ is an arbitrary distance metric and we use $L_2$ distance as $d$ in our experiments. For each $t$, we compute the hazard rate $h(t)$ as the sum of token logit probabilities $p(x_t = w | x_{<t}^*)$ for all forking tokens $w$ (i.e. tokens where $d(o_{t,w}, o_{t,w^*}) > \epsilon$).

## 3 EXPERIMENTS

We analyzed 7 unique tasks, across 4 different categories representative of typical LLM use and evaluation: Symbolic Reasoning, Mathematical Reasoning, Complex Question Answering, and Story Generation. These categories demonstrate the broad utility of our approach for various applications of text generation. The first three categories typically benefit from Chain-of-Thought (CoT) reasoning, where an LLM explicitly lists each step of its reasoning before giving a final answer (Kojima et al., 2022; Wei et al., 2022). This is appealing for our analysis since output text in CoT is more complex than a simple one-word answer, and so we may expect to see uncertainty dynamics in reasoning text. The category of Story Generation demonstrates the applicability of our methods to open-ended LLM

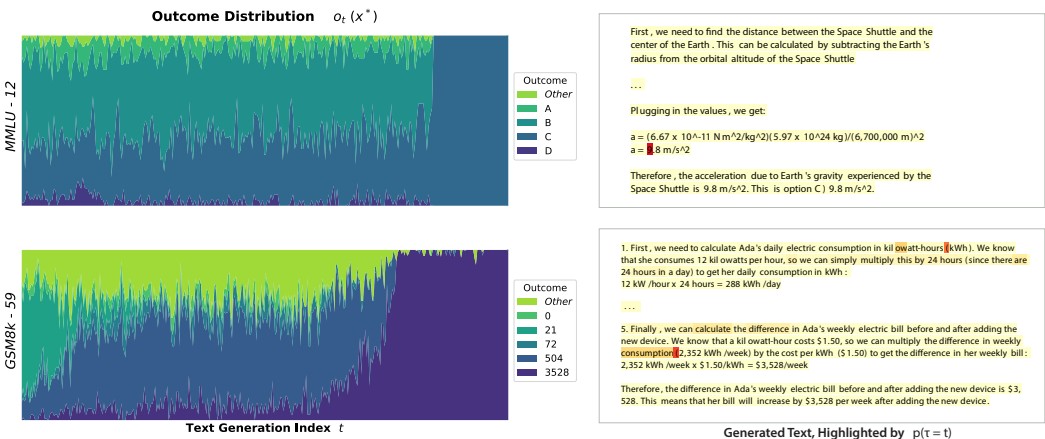

Figure 5: **Further examples of forking tokens** Two examples of outcome distributions $o_t(x^*)$ with forking tokens: a physics question from MMLU (Top; *Correct Answer: B*) and a mathematical reasoning question from GSM8k (Bottom; *Answer: 21*). In MMLU-12, we see a similar pattern to Fig. 4, where $o_t$ remains stable for most of the base path, before suddenly collapsing when the answer token is first specified *"a = 9.8 m/s ^ 2"*. In GSM8k-59 we see multiple forking tokens, both occurring at unexpected places. E.g. for the second change point, $o_{>t}$ changes depending on whether the token '*(*' or '*by*' is sampled. We also note that, similar to Fig. 4, at $t = 0$ the most probable outcome is the correct answer (*21*) but this answer disappears from $o_t$ part-way through.

use cases, such as creative writing, where there is no ground truth 'correct' answer. We have two tasks for each of the three CoT categories: one multiple choice dataset with a limited set of answers, and one dataset with free-text response answers. E.g. for Symbolic Reasoning, CoinFlip has two options for the answer (*"Yes"* or *"No"*), whereas LastLetter has no such constraint.

We use a zero-shot CoT prompt as in Kojima et al. (2022) for the first 6 tasks. CoinFlip (Wei et al., 2022) is a very simple symbolic reasoning task with two responses: *Yes* or *No*. LastLetter (Wei et al., 2022) is more complex symbolic reasoning task, prompting models to take the last letter of each of four names (e.g. *'Forrest Juanito Allan Candice'*) and concatenate them (*'tone'*). AQuA (Ling et al., 2017) and GSM8k (Cobbe et al., 2021) test mathematical reasoning with relatively simple math word problems. AQuA is 4-option multiple choice format, whereas GSM8k is open ended. MMLU (Hendrycks et al., 2020) complex question answering dataset of multiple choice questions spanning many domains and is used to test LM question-answering across a wide range of subjects. HotpotQA (Yang et al., 2018) is a complex question answering dataset of multi-hop reasoning questions which cannot be answered by a single memorized fact, but instead require chaining facts together. For our story generation task, we use the StoryCloze (Mostafazadeh et al., 2017) dataset, which was originally used for story understanding, and consists of short stories each with a valid ending sentence as well as an invalid ending. We modify StoryCloze for open-ended story generation, by prompting a model to generate a short story given only the first sentence of a scenario, with the constraint that the story must end with one of two provided endings.

For our Forking Paths Analysis, we sample the $k \leq 10$ most probable alternate tokens $x_t = w$ such that the probability of each token $w$ is at least 5%. When sampling batches at each token index and alternate token, we collect $S = 30$ text samples. For (1), we collect $N = 300$ full text responses $x$ from the starting index $t = 0$ and aggregating outcome responses $R$ into a histogram. For (2) we append a final answer prompt to $x^*$: *"... Therefore, the answer is: _"* and query the evaluated LLM, taking the logit probability of the first response token. (3) appends another prompt to the result of (2): *"... Percent confidence in final answer: _"*.

We evaluated OpenAI's GPT-3.5 completion model (gpt-3.5-turbo-instruct-0914; ~$2 per 1M tokens). For cost efficiency, we used Google's Gemini Flash (gemini-1.5-flash-001; ~$.075 per 1M tokens) to extract final answer outcome representations $R$. We used slightly different prompts for $R$ for each task, for example with MMLU we requested the answer choice *A/B/C/D* if it's provided. Additionally, we used simple answer cleansing functions written in Python (as in Kojima et al. (2022)) to extract minimal categorical answers from the $R$ model's responses. See App. G prompts and further details. Our Forking Paths Analysis pipeline (Fig. 2) is very costly in terms of the number of tokens required

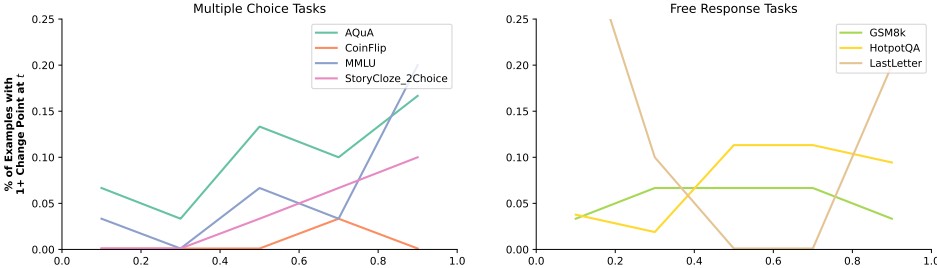

Figure 6: **Change points occur closer to the beginning of sequences for some tasks, and near the middle of sequences for others** Each point represents the fraction of question-answer examples in a task where our change point model predicts one or more change points approximately at $t$, i.e. where $p(\tau = t|y)$ is above a threshold of .7. In three tasks (AQuA, MMLU, and LastLetter) we find a large number of change points close to the end of responses, which may suggest patterns similar to MMLU-12 in Fig. 5.

for inference (~1 million tokens per $x^*$, or ~\$2 USD for GPT-3.5). For this reason we analyzed a subset of 30 data points for each task, ~\$500 in total. We randomly sampled question-answer pairs for all datasets. For GSM8k and MMLU, we used tinyBenchmarks (Polo et al., 2024), which are a subset of 100 examples where LLM performance on this subset is highly correlated with performance on the full dataset. For HotpotQA and AQuA we excluded questions and answers with string length outside the $[.1, .9]$ quantile range, and for HotpotQA we sampled ~18 data points for each difficulty level (easy, medium, hard).

## 4 RESULTS

### 4.1 CHANGE POINT DETECTION

In Fig. 4 we observe striking uncertainty dynamics in $o_t$ for HotpotQA (question 8076). The outcome distribution remains stable with the top single answer being *Robin Tunney* (the correct answer) until the forking token $t = 34$, when the distribution suddenly collapses to *Mia Sara* (a wrong answer). This is precisely when the LLM answers the first question in its chain-of-thought: *"First, we need to identify the American actress born on June 19, 1972 ... is Mia Sara."*. In reality, Mia Sara is an actress born on June 19 of a different year, 1967. Apparently the LLM was uncertain until it hallucinated a falsehood in its first reasoning step, and then it suddenly became certain that *Mia Sara* was the answer. We also see another hallucinated falsehood in the second reasoning step (Mia was not in this film), except by this point the LLM has already decided what its final answer will be.

In MMLU-12 we see a similar pattern (Fig. 5, Top), where $o_t$ remains stable in relative uncertainty for a long period, before suddenly collapsing when the answer token is first specified *"a = 9.8 m/s ^ 2"*. Note that the equation in the prior sentence *"a = 6.67 ... "* evaluates to a different answer *B) 8.9 m/s ^ 2*. Examining token logits we see that *9* had 54% probability of being generated and *8* had 29%. With GSM8k-59 we observe more complex uncertainty dynamics (Fig. 5, Bottom) with multiple change points. These forking tokens occur at unexpected, seemingly innocuous tokens: both are open parenthesis tokens which provide extra information not required for the final answer (i.e. *non-essential clauses* such as *'(kWh)'*). For the second forking token, *'... weekly consumption ( ... '*, the top answer in $o_{>t}$ is *'\$3,528'* if the token sampled is *'( '*, but changes to *'\$504'* if the token sampled is *'by'*. We also note that, similar to Fig. 4, at $t = 0$ the top probability answer is the correct answer (*'21'*) but this answer disappears from $o_t$ part-way through $x^*$. Finally, in most cases where our model detects no change points, we observe time series where the LLM remains confident in a single outcome, with no notable uncertainty dynamics throughout text generation. Visualizations for all examples $x^*$ and analyses in our dataset are available through an interactive dashboard online. [*] Additional analyses are provided in App. E.

We find additional patterns when aggregating CPD model inference results across tasks (Fig. 6, Tab. 1). Different tasks have different numbers of change points $m$, as well as different patterns in common change times $\tau$. We find more change points in some tasks, such as GSM8k and LastLetter, than others, with the fewest change points in CoinFlip. In order to compare change point times $\tau$,

---

[*]Interactive dashboard: `https://forking-paths.streamlit.app/`

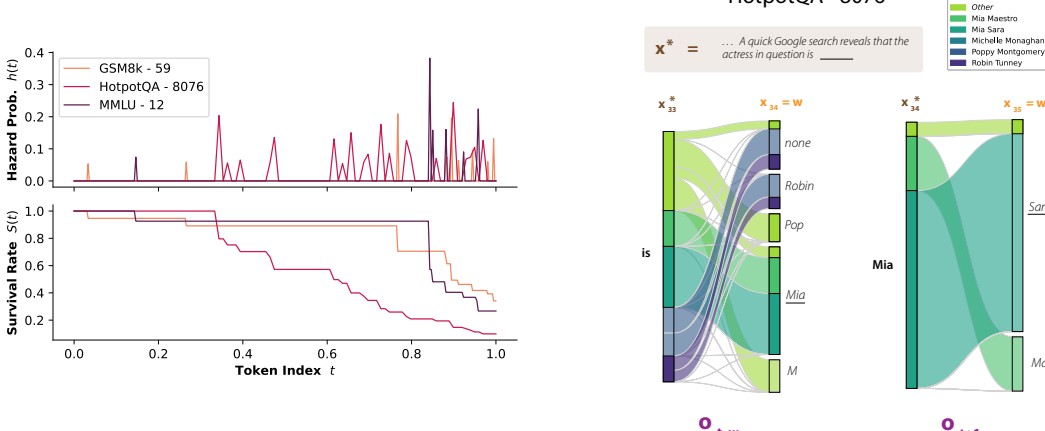

Figure 7: **Text generation has a low probability of surviving decoding without a major distribution shift** (Top Left) The hazard function $h(t)$ is the probability that $o_{t,w}$ will change significantly if a different token $w$ is sampled, and (Bottom Left) $S(t)$ measures the cumulative survival rate after many hazards. We see that in most cases, hazards slowly accumulate as $S(t)$ gradually decreases, but a few key tokens in GSM8k-59 and HotpotQA-8076 have large hazards corresponding to sharp drops in $S(t)$. We also see that the final survival rates $S(T)$ for all three examples are below 30%. (Right) $o_{t,w}$ visualized as parallel sets plots. We show distributions for two subsequent tokens $o_{t=34,w}$ and $o_{t=35,w}$, where different colors indicate different final answers. In this example, $t = 34$ and $t = 35$ are a step in a reasoning solution which eventually is used in the final answer (also see Fig. 4). For $t = 34$, we see that the next token being _Robin_ instead of _Mia_ will lead to completely different outcome distributions $o_{34,w='Mia'}$ and $o_{34,w='Robin'}$.

we first normalize token index $t$ to the range $[0, 1]$. Then, for each bucket across $t$, we compute the fraction of question-answer examples $o_t(x^*)$ with change point probability $p(\tau = t|y)$ above some threshold. Under this analysis (Fig. 6, Left), we find change points at different times for different tasks. In three tasks (AQuA, MMLU, and LastLetter) we find a large number of change points close to the end of responses, which may suggest patterns similar to MMLU-12 (Fig. 5, Top). In LastLetter we find the most change points early in sequences, and in HotpotQA, GSM8k, and AQuA we find the majority of change points in the middle of sequences. These points may correspond to individual CoT reasoning steps, similar to the change point in HotpotQA-8076 (Fig. 4).

## 4.2 SURVIVAL ANALYSIS

Our survival analysis shows low survival rates for many sequences $x^*$, even with a large distance threshold $\epsilon = .6$ (Fig. 7), using $L_2$ as the distance metric $d$. To give an intuition for $\epsilon = .5$ with $d = L_2$: a change from $o_{t,w^*} = [.5, .5]$ to $o_{t,w} = [.85, .15]$ will have a distance of $< .5$. The hazard probability $h(t)$ for the sequence with $\epsilon = .6$, with sharper spikes (and corresponding drops in survival rate $S(t)$) at some tokens more than others. This shows that even though even when forking tokens $w$, which radically change the outcome distribution, have a relatively low probability, these chances can accumulate over the course of generating a sequence. Aggregating results across all experimental data $x^*$ (Fig. 7, Right), we find that a majority of examples in our data have low survival rates at the end of the sequence ($S(T) < .2$) for all $\epsilon < .9$. This suggests that text generation may have a low probability of surviving decoding without a major distribution shift, which would imply that single sample LLM uncertainty estimates may be highly unstable.

## 5 DISCUSSION

Text generation with LLMs can be viewed as a branching tree of possible paths, where each word choice determines what text will follow, akin to Borges' _Garden of Forking Paths_ and other choose-your-own-adventure stories (Borges, 1941; Bottou & Schölkopf, 2023; janus, 2021). Many of these paths will follow similar trajectories and end in similar places, but some of them will hit forks which bifurcate into multiple distinct meanings. Our results support the Forking Tokens Hypothesis by empirically demonstrating forking tokens in a state-of-the-art LLM applied to various real world benchmarks, suggesting that LLMs are often just a single token away from producing

| Domain | Task | $m \geq 1$ **Changes** | **Mean** $S(T)$ |
|---|---|---|---|
| *Symbolic Reasoning* | CoinFlip | 0% | .20 |
| | LastLetter | 63% | .30 |
| *Mathematical Reasoning* | AQuA | 30% | .33 |
| | GSM8k | 27% | .18 |
| *Complex Question Answering* | MMLU | 43% | .13 |
| | HotpotQA | 32% | .26 |
| *Story Generation* | StoryCloze* | 7% | .27 |

Table 1: **Summary of results across all tasks** We used 7 tasks spanning 4 domains commonly used for LLM evaluation. For each task, we list results for both our Change Point Detection model and our Survival Analysis. For change points, we list the fraction of question/answer pairs in each dataset for which our model predicts at least 1 change point with 90% confidence (i.e. the .1 quantile of $p(m|y)$). We then list the average survival rate at the end of each sequence $S(T)$, using a threshold $\epsilon = .6$. We find lower survival rates in GSM8k and MMLU.

a very different answer. Forking Paths Analysis reveals dynamics unseen by prior approaches to uncertainty estimation, for example patterns where uncertainty is stable until a forking token is reached, at which point the outcome distribution collapses into certainty in a single answer. Our results show how static estimates of uncertainty can be misleading, e.g. estimating ~100% confidence at the last token and ~40% confidence at the first token of the same sequence (Fig. 4).

Forking tokens might also have important implications for LLM evaluation and safety. If LLM behavior can change suddenly when even one token is sampled differently, this could impact user safety, e.g. if an LLM suddenly shifts to the wrong distribution such as producing hallucinations or harmful language (Anil et al., 2024). Static safety evaluations may prove brittle when, in real world user interactions, users might accidentally or intentionally guide LLMs down dangerous paths. Static evaluations might be misleading measures of performance and alignment if LLMs have capabilities that emerge and then disappear over a single context window, or capabilities that remain dormant until a single token (or 'path') triggers them (Cleo Nardo, 2023; Li et al., 2024).

We see a number of promising avenues for future work. The most immediate directions would be further Forking Paths Analyses with different LLMs, including open-source, and new tasks. Our experiments use a second LLM as a one-hot feature extractor $R(\cdot)$, but more open-ended tasks could be analyzed using $R$ as a semantic vector embedding. A limitation of our approach is that it is very costly in the number of tokens sampled. More efficient sampling (Banga & Balsa-Canto, 2008) might be able to reduce the number of tokens needed, and it might even be possible to avoid sampling altogether if hidden activations can be used to predict forking. We also hope to further explore applications of Forking Paths Analysis, for example to improve process-level RL feedback (Lightman et al., 2023) or to guide branching in inference-time tree search (Yao et al., 2024). By studying the mechanisms of forking tokens, we might be able to better understand how LLMs represent uncertainty, or to steer models more effectively by patching activations at forking tokens (Fei et al., 2024).

The Forking Tokens Hypothesis raises a number of important questions, including: why do forking tokens occur in LLM text generation? One way to answer this question is with theories of In-Context Learning as Bayesian model selection (Xie et al., 2021). If this is the case, then we might expect sharp transitions in overall behavior when new data changes which latent model (i.e. 'capability' or 'concept') is the maximum a posteriori (Bigelow et al., 2024). Studying forking tokens may provide valuable insights into how model selection surfaces in LLMs operating in real-world domains. Another perspective on forking tokens is to consider whether humans might have similar phenomena. We might expect forking in human language comprehension, e.g. when you read the sentence *"Billy woke up in a _"*, the next word being either *hotel* or *hole* should change your expectations of the following words. In language production, however, a person who accidentally says the word *"hole"* instead of *"hotel"* is unlikely to then change their story to match *"hole"*. An LLM, on the other hand, might do just that. One interpretation might be that people typically holding intents and plan responses to some degree before they speak, whereas LLMs truly decide what to say next on the fly. This may be a fundamental property of next-word prediction models, unless the model has a hidden state such as a hidden chain-of-thought (OpenAI, 2024). Forking in human speech may be more common during certain kinds of creative dialog, e.g. when a person makes up a fictional story one sentence at a time, or when people think 'out loud' or 'step-by-step' (Lombrozo, 2024).

## REPRODUCIBILITY STATEMENT

All code and data used for this project is available at https://github.com/ebigelow/forking-paths.

## ACKNOWLEDGMENTS

E.B. was supported by a research internship with NTT Research, Inc. Compute resources used in this work were funded by a grant from the Hodgson Innovation Fund at Harvard's Department of Psychology. We would also like to thank Yang Xiang, Yingqiao Wang, and Ekdeep Singh Lubana for helpful conversations.

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

# Appendices

## A  RELATED WORK

- **Uncertainty estimation and calibration in LLMs**    Previous approaches to uncertainty estimation with LLMs have provided valuable insights (Geng et al., 2024; Xiong et al., 2024; Kadavath et al., 2022; Tian et al., 2023; Guo et al., 2017; Ye et al., 2024). However, prior re-sampling based uncertainty estimation does not effectively capture the space of forking paths, for example paths that are very likely to branch off of the highest probability (i.e. greedily decoded) branch. Final token probabilities or text-based uncertainty estimates (e.g. *"70%"*) likely do not capture the full picture. It may be possible to develop approaches to more effectively sample the space of possible paths that are simpler and/or cheaper than Forking Paths Analysis. Lastly, we find a parallel to our work in the area of conversation forecasting, which strives to estimate people's belief uncertainty over the course of a conversation (Sicilia et al., 2024).

- **Semantic diversity in text generation**    Semantic diversity (Tevet & Berant, 2021; Han et al., 2022; Kirk et al., 2023) measures the degree to which a language model generates meaningfully distinct responses to the same input. We believe semantic diversity may be a key cause of forking tokens, in that semantic diversity demands some degree of uncertainty in text generation. For example, diversity in CoT reasoning requires producing multiple distinct proofs.

- **Chain of thought and similar reasoning techniques**    Chain-of-Thought (CoT) reasoning (Kojima et al., 2022; Wei et al., 2022) and related techniques prompt an autoregressive language model to reason across the intermediate tokens it generates. One challenge in CoT reasoning is backtracking (Gandhi et al., 2024), where LLMs struggle to 'undo' a missed step. The Forking Tokens Hypothesis describes this phenomenon more broadly, where a single token can trigger distribution shift such as one reasoning path over another. On the other hand, LLMs are not always faithful to their chains of reasoning in the token stream (Turpin et al., 2024). Forking Paths Analysis may be able to shed further light on these cases.

## B   CHANGE POINT DETECTION MODEL DETAILS

We use an extremely efficient implementation of Bayesian multiple CPD, the Bayesian Estimator for Abrupt changes in Seasonality and Trends (BEAST). BEAST is described in Zhao (2019) and available as an R package at https://cran.r-project.org/web/packages/Rbeast/index.html. BEAST is implemented in C/C++, and we found it to be between $1 - 10$ thousand times faster than comparable packages for multiple CPD, most of which also did not support inference of $p(m|y)$.

We use BEAST to infer the posterior probability of a change point at each time $p(\tau = t|y)$ as well as posterior of the number of change points in a time series $p(m|y)$. To estimate these posteriors, BEAST iteratively draws Monte Carlo samples $j$ for each of the following variables, in order: the number of change points $m \sim p(m^{(j)}|\sigma^{(j-1)}, y)$, change times $\tau \sim p(\tau^{(j)}|m^{(j)}, \sigma^{(j-1)}, y)$, segment parameters $p\beta, \delta \sim (\beta^{(j)}, \delta^{(j)}|\tau^{(j)}, \sigma^{(j-1)}, y)$, and noise parameter $\sigma \sim p(\sigma^{(j)}|\beta^{(j)}, \delta^{(j)}, \tau^{(j)}, y)$. We show plate notation for the structure of the BEAST CPD model in Fig. 8.

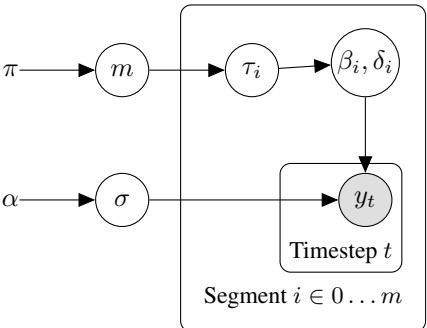

Figure 8: **Plate diagram for change point model**     $y$ is a univariate time series, $m$ is an integer number of change points, $\tau_i \in \{1 \ldots |y|\}$ is the time index when a change point occurs, $\theta_i$ includes model parameters such as abrupt change magnitude and polynomial trend model coefficients for segment $\hat{y} = f_{\beta_i, \delta_i}(y_{\tau_{i-1}:\tau_i})$ (note: there are $m + 1$ segments given $m$ change points), $\sigma$ is controls the variance of $y_t \sim Normal(\hat{y}, \sigma)$, $\pi(m)$ is a hyper-prior over number of change points $m$, and $\alpha$ is hyper-prior for noise $\sigma$.

As described in Sec. 2.3, a CPD model decomposes a time series $y$ into a set of $m$ segments, and each segment $i \in \{0 \ldots m\}$ is fit by a regression model with intercept (i.e. abrupt change) $\delta_i$ and slope $\beta_i$, applied to time steps $t$ between $\tau_{i-1}$ and $\tau_i$. In our case, we assume linear models for each segment $y_t = \beta_i t + \delta_i$ , $t \in \{\tau_{i-1}, \ldots, \tau_i\}$, to match our assumption and qualitative observation that in $o_t$ there are stable regimes of uncertainty which continue for many tokens, until $o_t$ abruptly changes to a new distribution. We also observe 'drift' in some cases, where $o_t$ slowly changes from one distribution to another, which in our model corresponds to different values of $\beta_i$. To our knowledge, our work is the first apply CPD to analyze neural network learning dynamics, either in-context (as in our case) or in-weights. (Hu et al., 2023) uses Hidden Markov Models to analyze in-weights learning dynamics, which achieves a similar purpose as CPD [*], but with less interpretable parameters. We see an exciting direction for future work being to further understand which statistical modeling assumptions are most appropriate for describing uncertainty dynamics in text generation.

One challenge we found with using BEAST for CPD is a high false positive rate in cases where $o_t$ has fewer changes. In these cases, the drift $y_t$ has a low magnitude overall, and so very small changes in $y_t$ can show up as false positive change points when BEAST re-normalizes $y_t$. To address this, we manually tuned noise hyper-parameter $\alpha$ and slightly perturb $y_t$ with Gaussian noise of variance .03.

In our CPD and survival analysis models, we used $L_2$ distance. We also tested with $L_1$ distance and K-L divergence, but found that results with $d = L_2$ most reliably corresponded to qualitative judgments of change points in $o_t$ and $o_{t,w}$.

---

[*]HMMs can be used for change point detection, as in Luong et al. (2012)

## C COMPARING UNCERTAINTY ESTIMATION METHODS

### C.1 STATIC UNCERTAINTY ESTIMATES

We now compare out outcome distribution representation $o_t$ to three *static* uncertainty estimation baselines, inspired by work such as Xiong et al. (2024). (1) We estimate the outcome distribution by re-sampling $N = 300$ completions from the first token $t = 0$ and on. (2) We take the base path $x^*$, append a brief string to the end *Therefore, the answer is: ___*, and we take the logit probabilities for the next token as the answer certainty estimate. (3) Given the model's output (greedily decoded tokens) in (2), we then prompt the model for its confidence by appending an additional prompt *Percent confidence in final answer: ___*. For (3), we take the numeric % confidence estimate and assign that confidence to the greedy token output in (2), and all other confidence to a generic *'Other'* outcome.

We see in Figs. 9, 10 that in these cases with complex uncertainty dynamics and change points, the static confidence estimate (1) is significantly different from (2) and (3). This is easily explained by looking at $o_t$, since the outcome distribution at the beginning of the sequence $o_0$ matches the first baseline (1), and the outcome at the end of the sequence $o_T$ approximately matches (2) and (3). We also find that confidence estimates (2) and (3) assign very high certainty to a final answer, despite there being substantial fluctuations in uncertainty over the course of text generation.

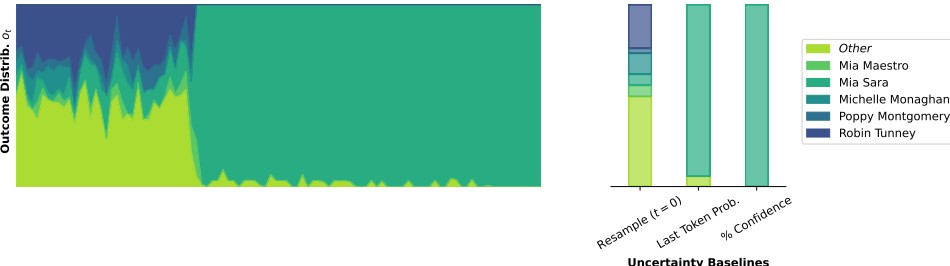

Figure 9: **Comparing static uncertainty baselines to Forking Paths analysis (HotpotQA-8076; Fig. 4)** The uncertainty dynamics we observe in $o_t$ (Left) are invisible to static uncertainty estimate baselines (1) - (3) (Right). We also observe that baseline (1) is different from (2) and (3), with the top answer in (1) being *Robin Tunney* (the correct answer), whereas (2) and (3) assign near-100% certainty to *Mia Sara*.

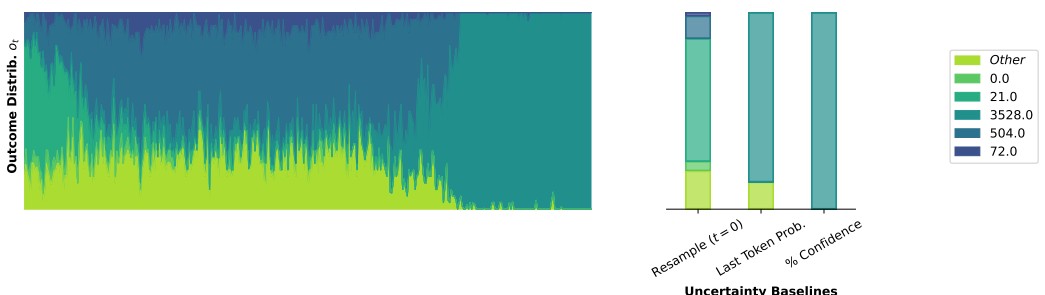

Figure 10: **Comparing static uncertainty baselines to Forking Paths analysis (GSM8k-59; Fig. 5)** We see similar patterns here as in Fig. 9, with uncertainty dynamics in $o_t$ (Left) that are hidden behind static uncertainty estimates (Right). In this case, (1) assigns the majority of certainty to the correct answer *21*, where as (2) and (3) are near-100% certainty in a different answer *3,528*.

### C.2 TOKEN LOGIT PROBABILITIES

A simple question one might ask about forking tokens, is whether these can simply be explained as low-probability tokens which were unlikely, and when sampled caused the model to go off course.

For this reason we ran a correlation between the change point probability at a given token $p(\tau = t \mid y)$ and the token logit probability $p(x_t = w^*)$ for the greedy token $w^*$. As shown in Fig. 11, we find minimal correlation between (log) change point probability and (log) token probability. The slight correlation we find is positive, contrary to the question above, and we note that many of the highest-probability forking tokens also have high logit probabilities.

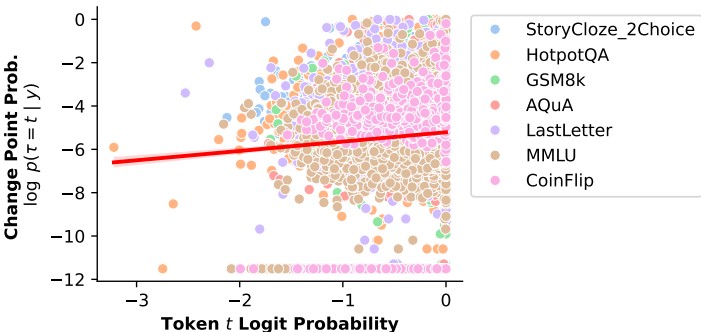

Figure 11: **Correlation between change point probability and token logit probability** We find that token logit probability $p(x_t = w^*)$ is not strongly predictive of the probability that a token is labeled a change point by our model $p(\tau = t \mid y)$. In fact we find a slight positive correlation, and that the highest probability change points also have high token logit probabilities.

### C.3 COMPARING CPD AND SURVIVAL ANALYSIS

Another simple question is how our two analysis methods, change point detection (Sec 2.3) and survival analysis (Sec 2.4), compare to one another. If our methods make similar predictions about which sequences have forking tokens, they might be redundant. In Sec. 2.4 we explain why survival analysis of $o_{t,w}$ may provide different results from change point detection. We test this by running a correlation between the estimated number of change points predicted by out change point model (i.e. the .1 quantile of $p(m \mid y)$) and the final survival rate $S(T)$ of a sequence. As shown in Fig. 12, we find $\approx 0$ correlation between which samples have low survival rates with which samples have more change points. This suggests that our two methods are identifying different forking tokens in the outcome distributions $o_t$ and $o_{t,w}$. Though $o_t$ and our change point detection models are given emphasis in the present work, further analysis of $o_{t,w}$ may also be a promising direction for future work.

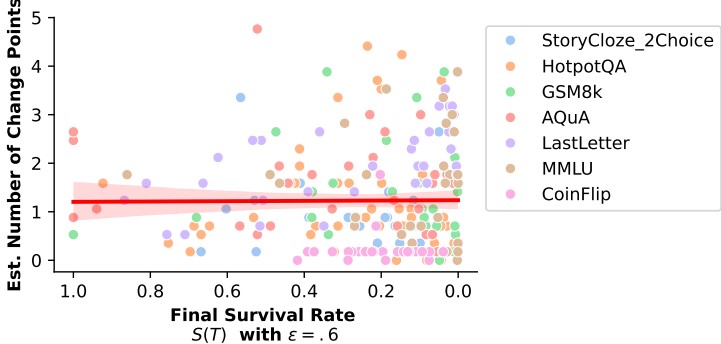

Figure 12: **Correlation between change point model and survival analysis.** We find no correlation between the number of change points estimated by our CPD model (the .1 quantile of $p(m \mid y)$) and the final survival rate $S(T)$ of a sequence. Each point represents a single prompt and base path $x^*$.

# D IMPROVING COMPUTATIONAL EFFICIENCY

As mentioned in Sec. 3, the main limitation of Forking Paths Analysis is that it is very costly in terms of number of tokens sampled. The approach we used has the token complexity: $O\big((|x_{in}| + |x^*| + |x^{(s)}|) * |x_t = w| * |x^*| * S\big)$, where $|x_{in}|$ is the input prompt, $|x^*|$ is the length of the base path, $|x^{(s)}|$ is the length of output completions, $|x_t = w|$ is the number of alternate tokens at an index $t$, and $S$ is the number of completions sampled for each of these.

For our experiments, we used $S = 30$ and sampled on the order of millions of tokens for each input and base path. However, one question we asked was whether a smaller number of samples might serve nearly as well to identify forking tokens. As shown in Fig. 13, we find that with 10-20 samples, the number of change points our CPD model predicts is very similar to when we use $S = 30$. In other words, our experiments could be run at half the cost and with similar results.

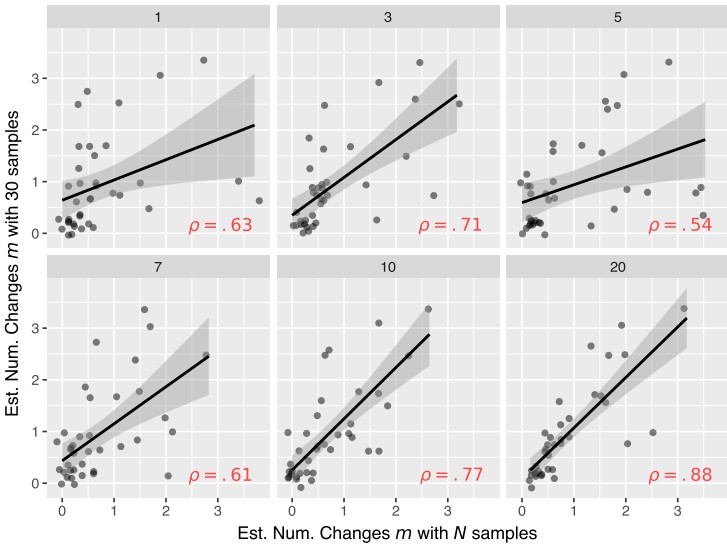

Figure 13: **Correlation between number of change points estimated with our full dataset ($S = 30$) and estimated with smaller sample sizes $S = N$**    For each panel, we sub-sampled $N$ completion texts for each token $t$ and $w$, where $N < 30$. Given this smaller dataset and estimated outcome distribution $o_t$, we predict the number of change points (.1 quantile of $p(m \mid y)$). In red, we plot Spearman's $\rho$ correlation coefficient.

Additionally, we see a number of avenues for future work to improve the efficiency of Forking Paths Analysis. By using prompt caching with open-source models [*], the token sample complexity may be reduced to $O\big((|x^*| + |x^{(s)}|) * |x_t = w| * |x^*| * S\big)$ (i.e. samples will not scale by $x_{in}$). Next, it may be possible to use statistical models to determine optimal tokens $t$ and $w$ to draw samples for. This is very similar to the problem of Optimal Experiment Design (Banga & Balsa-Canto, 2008), which uses statistical models to determine which data should be collected to most efficiently test a hypothesis. More ambitiously, with open-source models we may be able to use hidden activations to predict forking tokens. Specifically, we can test whether hidden activations can predict token model predictions $p(\tau = t \mid y)$, $p(m \mid y)$ (for our CPD model), and $S(T)$ (for our survival analysis). If this is possible, it may be possible to avoid the costly token sampling altogether, simply by analyzing model activations.

---

[*]e.g. https://huggingface.co/docs/transformers/en/kv_cache

# E  ADDITIONAL ANALYSES

Visualizations for all examples $x^*$ in our dataset and their respective analyses are available online through an interactive dashboard: https://forking-paths.streamlit.app/.

Below, we include a subset of examples, and show $o_t$, $p(\tau = t|y)$, and highlighted text for each example $x^*$. Refer to Fig. 4 for how to interpret figures. Examples are hand-selected to demonstrate interesting uncertainty dynamics, including change points. However, we also found many other interesting examples not shown here.

COINFLIP

In the CoinFlip task (Fig. 15), most outcome distributions are static over the course of text generation. This task is particularly easy for GPT-3.5, and from $o_t$ we conclude that, from the beginning of text generation, the LLM 'decides' for certain what its final response will be.

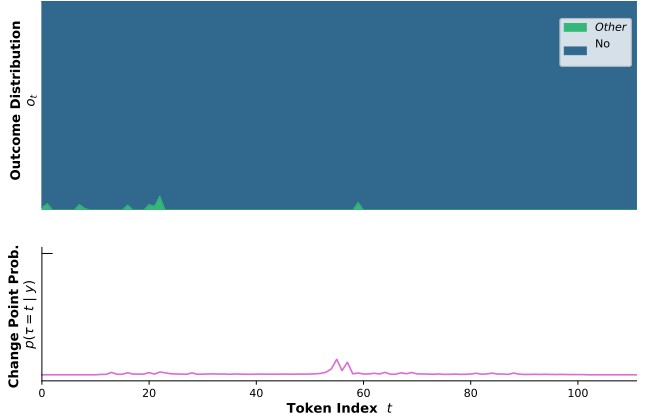

Question:
A coin is heads up. Deshon flips the coin over. Jamariea does not flip the coin over. Is the coin still heads up?

Answer:
Let's think step by step. Initially, the coin is heads up. Then, Deshon flips the coin over, which means he turns the coin to the other side. So now, the coin is tails up. However, Jamariea does not flip the coin over, which means she does not change the position of the coin. Therefore, the coin is still tails up. It is no longer heads up because Deshon flipped it over, but it is not heads up either because Jamariea did not flip it back. So, the coin is not heads up anymore.

Figure 14: **CoinFlip-1035**

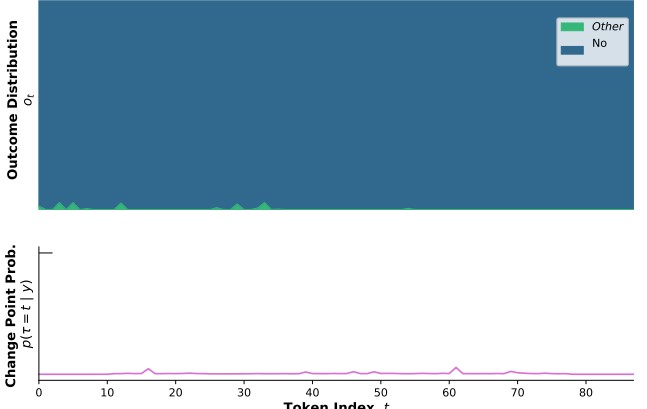

Question:
A coin is heads up. Aalisha flips the coin over. Cleotha does not flip the coin over. Is the coin still heads up?

Answer:
Let's think step by step. Initially, the coin is heads up. Aalisha flips the coin over, which means she turns the coin over so that the other side is facing up. This means that the coin is now tails up. However, Cleotha does not flip the coin over, so the coin remains in the same position as Aalisha left it - tails up. Therefore, the coin is no longer heads up, it is now tails up.

Figure 15: **CoinFlip-15426**

LASTLETTER

In the LastLetter task (Figs. 16, 17), we observe more change points than any other tasks. Many of these follow a very consistent pattern: the outcome distribution $o_t(x^*)$ remains uncertain until the final answer tokens, at which point it collapses to a single outcome.

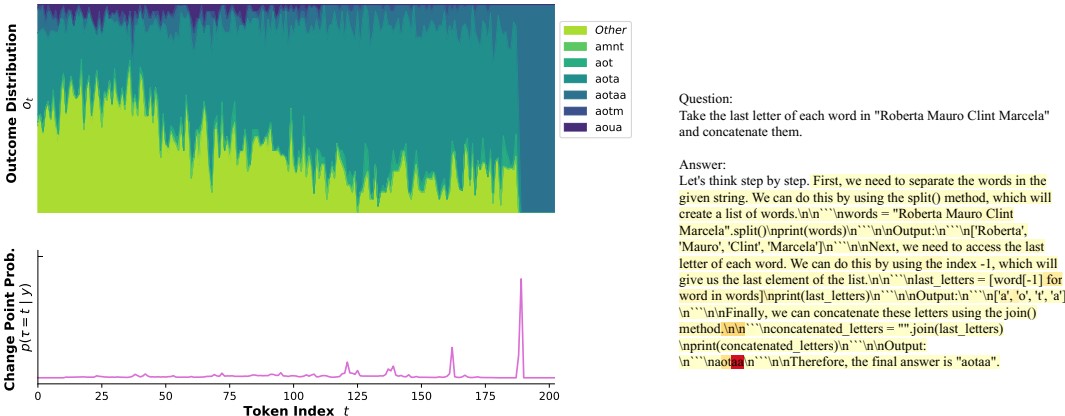

Figure 16: **LastLetter-103** – Correct answer: *aota*

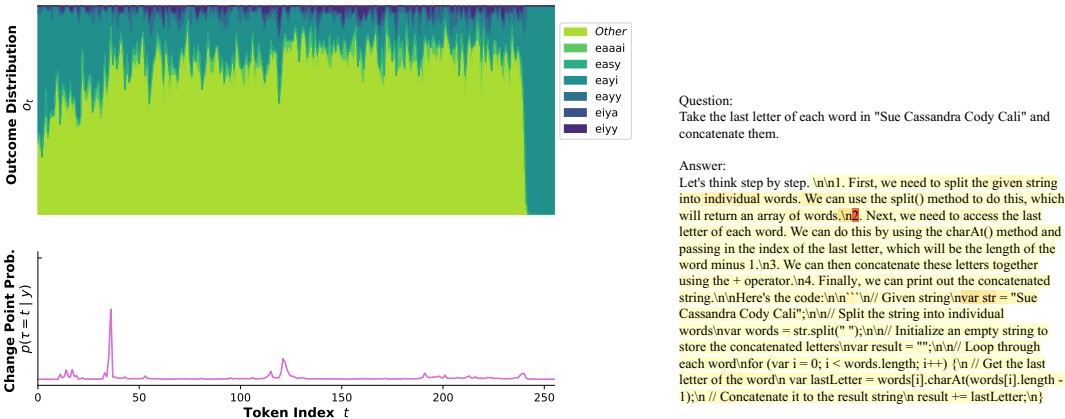

Figure 17: **LastLetter-220** – Correct answer: *eayi*

AQuA

We find the most complex uncertainty dynamics in the Mathematical Reasoning domains, AQuA (Figs. 18, 19) and GSM8k (Figs. 20, 21).

For the AQuA examples shown here, we observe multiple changes, including changes at relatively unexpected tokens. In these examples, we also observe sharp changes which occur over the course of a few tokens instead of a single token, e.g. the second change in AQuA-62 and the first change in AQuA-160.

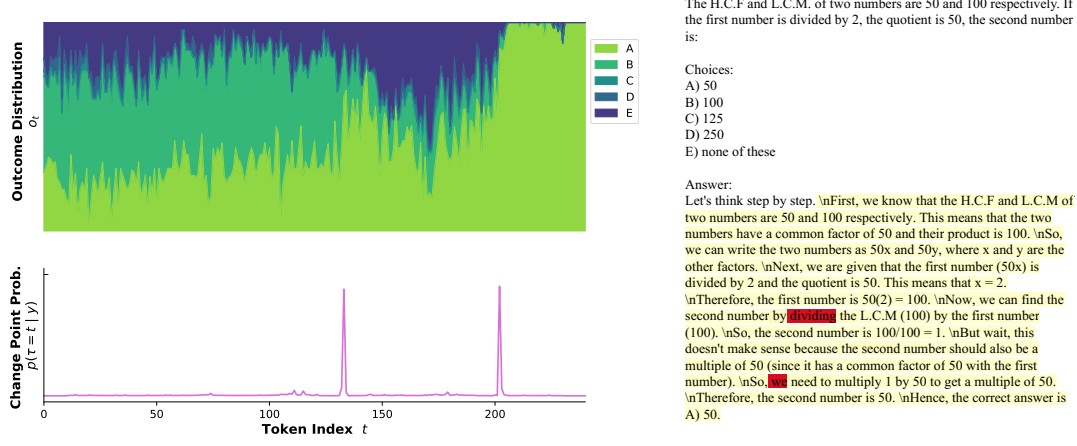

Figure 18: **AQuA-62** – Correct answer: A

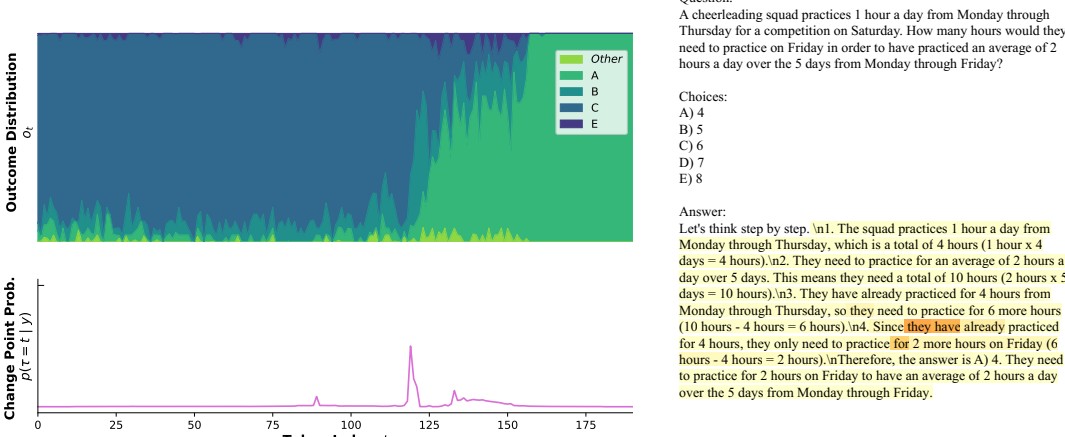

Figure 19: **AQuA-160** – Correct answer: C

GSM8K

For GSM8k (Figs. 20, 21), we similarly find complex uncertainty dynamics over text generation.

For the AQuA examples shown here, we observe multiple changes, including changes at relatively unexpected tokens. In these examples, we also observe sharp changes which occur over the course of a few tokens instead of a single token, e.g. the second change in AQuA-62 and the first change in AQuA-160.

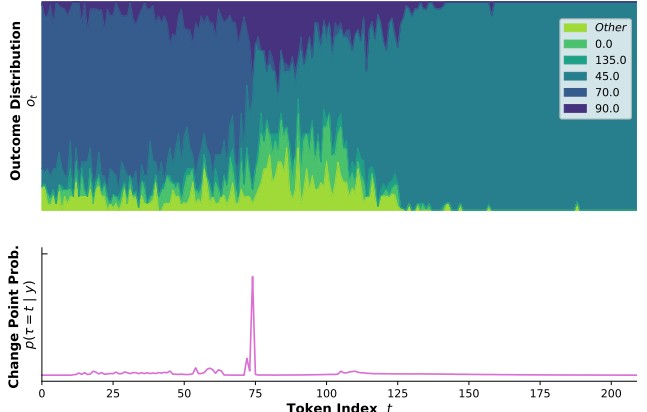

Figure 20: **GSM8k-1** – Correct answer: *70*

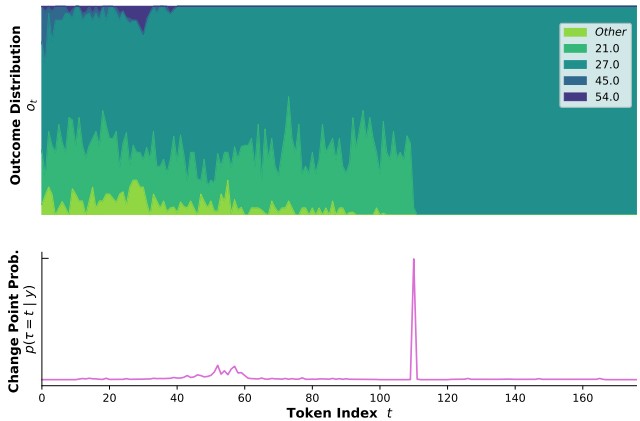

Figure 21: **GSM8k-78** – Correct answer: *54*

HotpotQA

We find dramatic uncertainty dynamics with change points in the HotpotQA examples shown in Figs. 4, 22. However, in HotpotQA we also observe a cases where different nearly identical outcomes are expressed with different words, as in Fig 23. While we tried to control for semantic variation by using a powerful LLM for $R(\cdot)$, gemini-1.5-flash-001, we see this as a general challenge with properly evaluating LLM performance on open-ended benchmarks. Curiously, in some of these cases such as Fig 23 we nonetheless observe stable and interesting uncertainty dynamics across outcomes that are only superficially distinct.

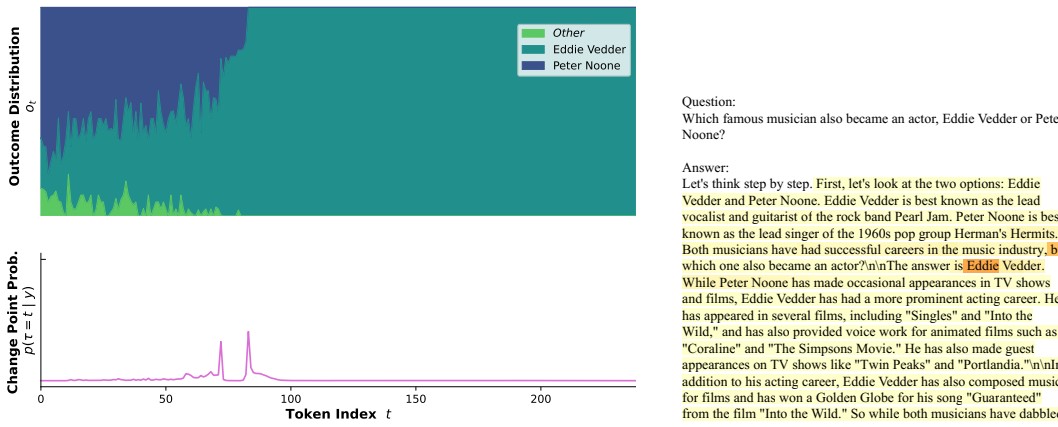

Figure 22: **HotpotQA-79442** – Correct answer: *Peter Noone*

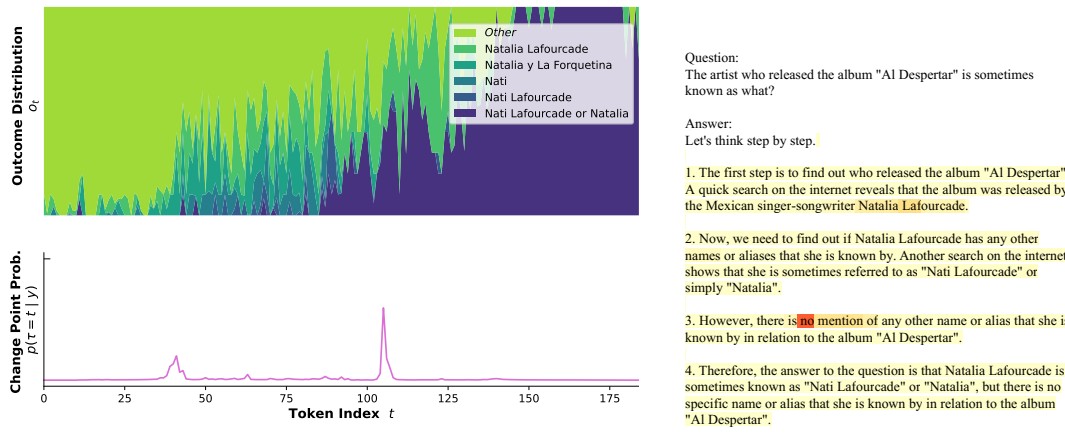

Figure 23: **HotpotQA-30010** – Correct answer: *La Negra*

MMLU

Sequences in the MMLU have the second most change points (Figs 24, 25, 26), with LastLetter having the most.

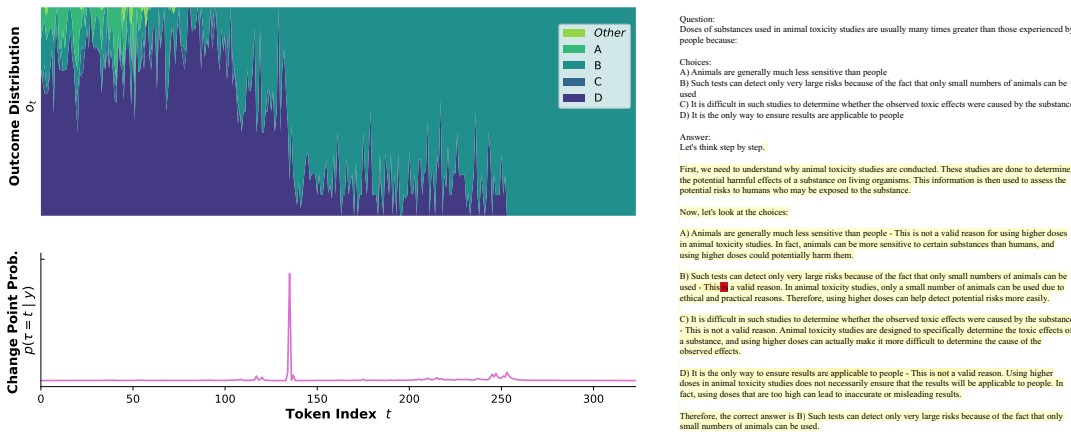

Figure 24: **MMLU-3** – Correct answer: *B*

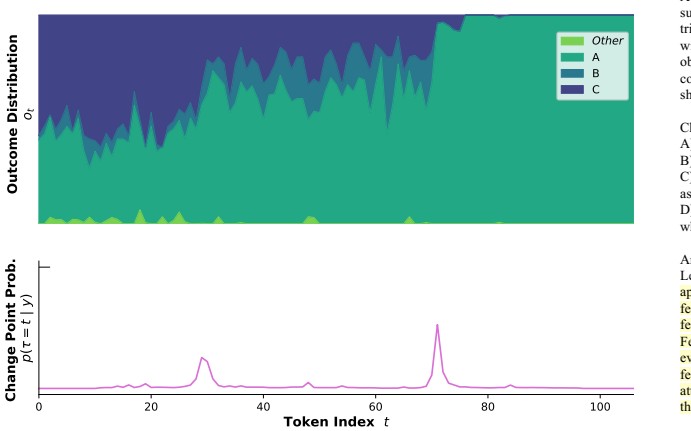

Figure 25: **MMLU-58** – Correct answer: *A*

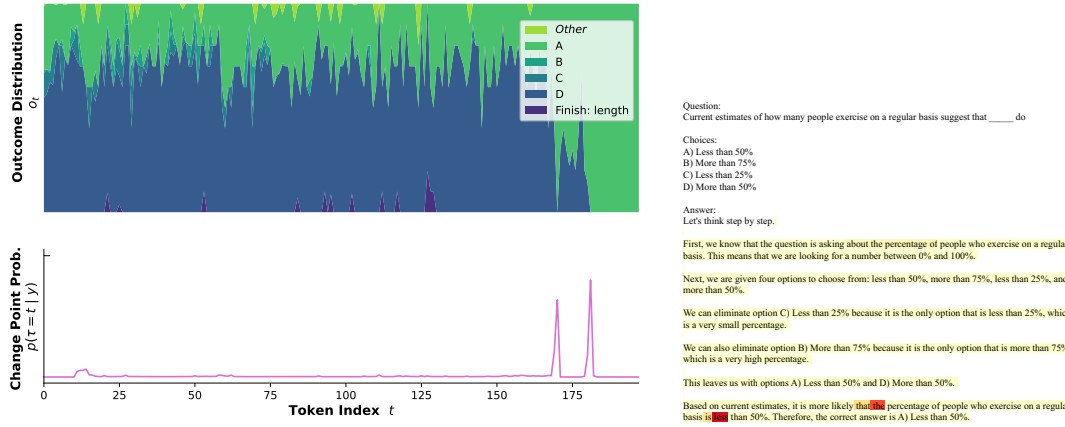

Figure 26: **MMLU-72** – Correct answer: *A*

STORYCLOZE

For StoryCloze, in some cases we observe change points, such as in Fig. 27. However, we also observe many cases (such as Fig. 28) of a different pattern, where $o_t$ gradually drifts from one distribution to another. This pattern is significantly more prevalent in StoryCloze than the other tasks we evaluated, which is also noteworthy since this is the only open-ended task which has no ground truth answer.

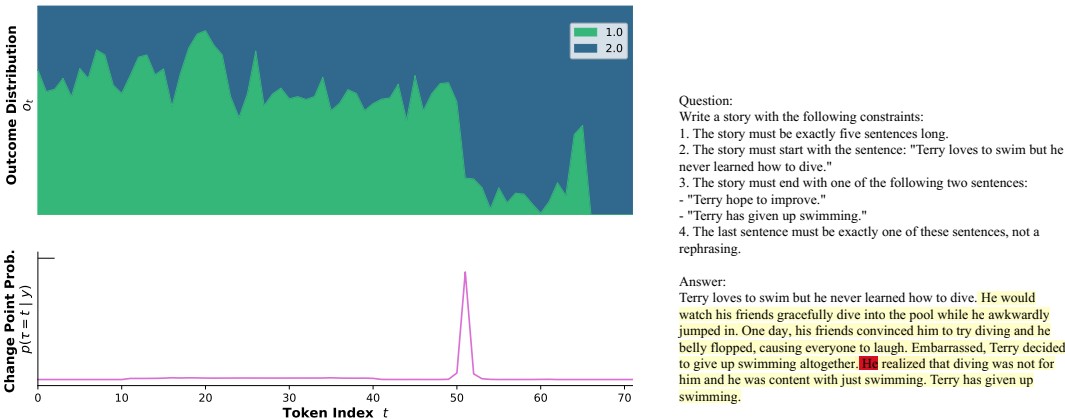

Figure 27: **StoryCloze-1197**

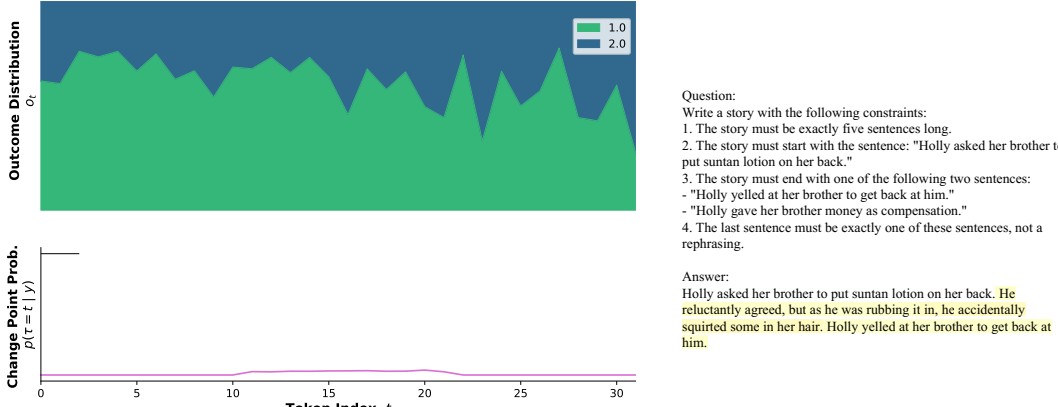

Figure 28: **StoryCloze-1482**

AGGREGATED ANALYSIS FOR NUM. CHANGE POINTS

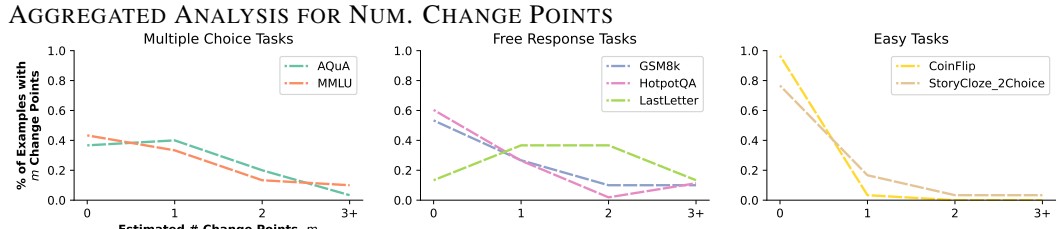

Figure 29: **Estimated number of change points** $m$ **aggregated over each task** We estimate the number of change points in each task by taking the .1 quantile of $p(m|y)$ (rounded to the nearest integer) for each prompt and base path $x^*$. We then compute the fraction of all examples in a task with each estimated number of change points.

In Fig 29, we show aggregate results for each task, estimating the number of change points. Intuitively, this serves as aggregating the posterior $p(m|y)$ over all data points $y$ in a single task. These results are computed by taking a single point estimate for the number of change points $m$, which in Fig 29 is the .1 quantile of $p(m|y)$. Note that the .1 quantile is equivalent to a Bayes factor of 9 between $p(m \geq m'|y)/(m < m'|y)$, where $m'$ is the estimate for $m$, similar to our hypothesis testing method described in Sec. 2.3.

# F    ANALYSES WITH VARYING THRESHOLDS

In Fig. 30, we show results for Figs. 6, 29 with varying threshold parameters. In the case of the change point time (Left plots), the threshold is used to convert $p(\tau = t|m)$ into a binary indicator for whether the change point probability is above some threshold. For the number of change points (Right plts), the threshold we vary is the quantile of $p(m|y)$ used to compute a single point estimate for the number of change points in a sequence $y$.

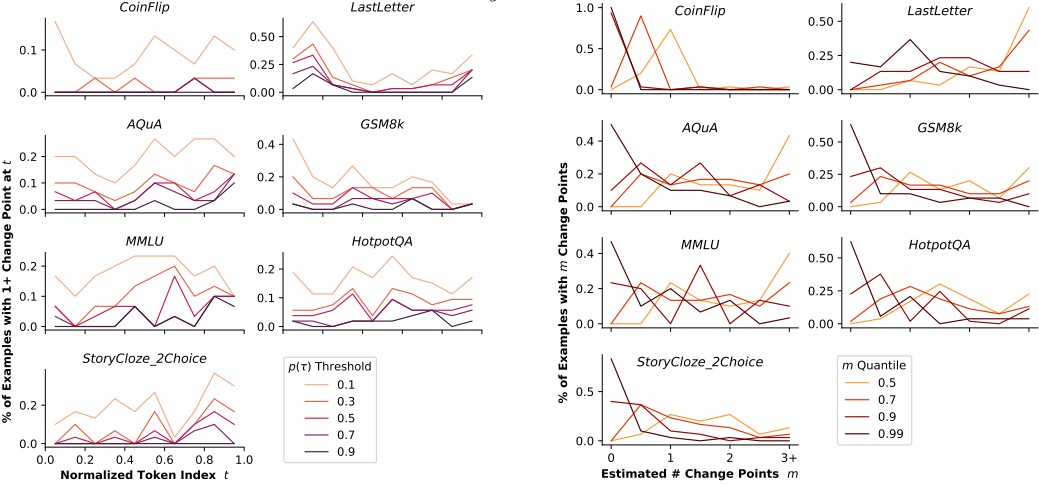

Figure 30: **Task-level estimates of CPD model results, varying threshold levels**    Here we show the results of Figs. 6, 29 while varying the thresholds we use for each. (Left) Varying the change point probability threshold for $p(\tau = t|m)$, and (Right) varying the quantile used to estimate $m$

## G    EXPERIMENT DETAILS

This sections lists the specific prompts used for completions, prompts for collecting outcome representations $R(\cdot)$, and finally answer cleansing functions used to parse outcomes into minimal answers.

### COMPLETION PROMPTS

Below are prompts used for sampling the base path $x^*$ as well as completions $x_t^{(s)}$ $\forall t, s$ used to for Forking Paths Analysis.

---

**Standard CoT Prompt (LastLetter, GSM8k, HotpotQA)**

Question: {question}

Answer: Let's think step by step.

---

**Multiple Choice CoT Prompt (AQuA [5 choice], MMLU [4 choice]**

Question: {question}

Choices:
A) {A}
B) {B}
C) {C}
D) {D}
E) {E}

Answer: Let's think step by step.

---

**StoryCloze 2-Choice Prompt**

Question:
Write a story with the following constraints:
1. The story must be exactly five sentences long.
2. The story must start with the sentence: "{ first sentence } "
3. The story must end with one of the following two sentences:
- "{ last sentence 1 } "
- "{ last sentence 2 } "
4. The last sentence must be exactly one of these sentences, not a rephrasing.

Answer: { first sentence }

---

### OUTCOME REPRESENTATION PROMPTS

The following prompts are used for extracting outcome representations $R(\cdot)$ from a second LLM. In our case, for cost efficiency the second LLM uses the ChatCompletions API format (messages dictionaries) instead of Completions (single text string).

**Yes/No Outcome Prompt (CoinFlip)**

```
[{
    'role': 'user',
    'content': < full_qa_text >
}, {
    'role': 'user',
    'content': 'What is the final choice (Yes or No) in the
    ↪  Answer in the previous message?'
}, {
    'role': 'system',
    'content': 'Respond with a single-word Yes or No if
    ↪  possible.'
}]
```

**Generic QA Outcome Prompt (LastLetter, HotPotQA)**

```
[{
    'role': 'user',
    'content': < full_qa_text >
}, {
    'role': 'user',
    'content': 'What is the final answer to the Question
    ↪  given in the Answer in the previous message? Be
    ↪  brief.'
}, {
    'role': 'system',
    'content': 'Respond with only the final answer, if
    ↪  possible. Be brief in your response, do not include
    ↪  unnecessary text.'
}]
```

**Multiple Choice Outcome Prompt (AQuA, MMLU)**

```
[{
    'role': 'user',
    'content': full_qa_text
}, {
    'role': 'user',
    'content': 'What is the final choice (A, B, C, or D) at
    ↪  the end of the Answer in the previous message?'
}, {
    'role': 'system',
    'content': 'Respond with a single-word multiple choice
    ↪  answer if possible: A, B, C or D.'
}]
```
\* Note: AQuA prompts instead specify A,B,C,D,E

---

**Numeric Outcome Prompt (GSM8k)**

```
[{
    'role': 'user',
    'content': < full_qa_text >
}, {
    'role': 'user',
    'content': 'What is the final answer given in the Answer
    ↪  in the previous message?'
}, {
    'role': 'system',
    'content': 'Respond only with a number if possible. Do
    ↪  not include units such as "$".'
}]
```

---

**Story Ending Outcome Prompt ( StoryCloze* )**

```
[{
    'role': 'user',
    'content': < full_qa_text >
}, {
    'role': 'user',
    'content': 'Which of the following two sentences matches
    ↪  the ending of this story?'  \
                f'\n1. " < last sentence 1 > "'  \
                f'\n2. " < last sentence 2 > "'
}, {
    'role': 'system',
    'content': 'Respond with a single word, either 1 or 2.'
}]
```

---

ANSWER CLEANSING FUNCTIONS

Here we list the 'answer cleansing' functions that we used to parse final answers from the extracted outcome representations $R(\cdot)$. Even with the structured outcome prompts listed in the prior section, the LLMs we used for $R(\cdot)$ have a tendency to occasionally respond verbosely even when instructed otherwise. For this reason, we design simple functions to extract, e.g. numeric values from GSM8k responses are parsed so that equal values are represented equivalently (e.g. $R(x_1) =$ "1.0" and $R(x_2) =$ "1"). This method is adopted from Kojima et al. (2022) (Appendix A.6), and the numeric and multiple choice answer cleansing functions we used are modified versions of their functions.

Also note that we label all outcomes which cannot be parsed, or which are outside top 6 most probable in $o_t$, as the value '*Other*'.

---

**Standard Answer Cleansing Function**

```python
def base_ans_fn(s):
    s = (s.split('answer is ')[1].replace('$', '')
        if 'answer is ' in s else s)
    s = s.strip()
    if len(s) <= 1:
        return s
    if s[-1] == '.':
        s = s[:-1]
    return s
```

---

**Multiple Choice Answer Cleansing Functions**

```python
def abcd_fn(s, other_tok='*Other'):
    s = s.strip()
    return s if s in ('A', 'B', 'C', 'D') else other_tok

def abcde_fn(s, other_tok='*Other'):
    s = s.strip()
    return s if s in ('A', 'B', 'C', 'D', 'E') else other_tok
```

**Numeric Answer Cleansing Function**

```python
import re

def numeric_fn(s):
    pred = s
    pred = pred.replace(",", "")
    pred = [s for s in re.findall(r'-?\d+\.?\d*', pred)]

    if len(pred) > 0:
        return str(float(pred[0]))
    return s
```

**Answer Cleansing Function for LastLetter Task**

```python
def last_letter_ans_fn(s):
    s = s.lower()
    s = s.replace('"', '')
    s = s.replace("'", '')
    s = s.replace('.', '')
    if 'answer is' in s:
        s = s.split('answer is')[1]
    if 'message is' in s:
        s = s.split('message is')[1]

    s = s.replace(' ', '')
    return s.lower().strip()
```

