# OpenReview forum: "Forking Paths in Neural Text Generation"
_ICLR.cc/2025/Conference — ICLR 2025 Poster_

### Official Review · Reviewer_5yCK · 2024-10-17

**Soundness:** 3
**Presentation:** 2
**Contribution:** 3
**Rating:** 6
**Confidence:** 2

**Summary:**

This article innovatively proposes to analyze the uncertainty of language model generation by analyzing the intermediate steps of LLM decoding. Around the defined forking tokens hypothesis, the authors have built a pipeline and designed CPD to study the uncertainty behavior of language models in different inference tasks. The experimental results demonstrate some very interesting conclusions and provide a comprehensive discussion on the impact of this method on future work.

**Strengths:**

* The assumption in this article is quite important, and the pipeline constructed to validate the assumption and the evaluation metrics are very interesting
* The experimental analysis in this article is detailed and comprehensive.
* The assumption in this article is very interesting and significant.

**Weaknesses:**

The evaluation of the uncertainty of language models designed in this article requires sampling a large number of generated results for different tokens and conducting evaluation analysis. Therefore, the cost of evaluating a single sample is also enormous, which may affect the scalability of this work. However, this does not negate the innovativeness of this work.

**Questions:**

No question

---

> ### Author Response · Authors · 2024-11-22
>
> We are glad the Reviewer finds our hypothesis and methods “important”, “very interesting”, and “significant”, and our experiments to be “detailed and comprehensive”.
>
>
>
> > The evaluation of the uncertainty of language models designed in this article requires sampling a large number of generated results for different tokens and conducting evaluation analysis. Therefore, the cost of evaluating a single sample is also enormous, which may affect the scalability of this work. However, this does not negate the innovativeness of this work.
>
>
> Indeed, we agree that this is a limitation of our current approach. However, we would like to emphasize that studying forking tokens and uncertainty dynamics in text generation is a completely new approach with no prior work. We see enormous opportunities for future work to improve on the efficiency of our methods.
>
> We have added an Appendix (D) which quantifies the computational complexity of our method, along with a number of suggestions for future work to reduce the cost of our analysis method. The most simple of these approaches would be to collect fewer text completion samples. To test this approach, we added an experiment in Appendix D which compares how many samples are needed to get reliable estimates of outcome distributions and forking tokens. This experiment suggests that reliable estimates might be obtained with ~½ the number of samples we use, at approximately ½ our total cost.
>
>
>
> **Rebuttal summary**
>
> We are grateful to the Reviewer for describing our work as “important”, “very interesting”, “innovative”, and “significant”, and our experiments to be “detailed and comprehensive”.
>
> The only weakness noted by the Reviewer is the computational cost of our sampling method. While we agree this is an important limitation for future applications, we consider the problem of improving efficiency as an exciting topic for future work, particularly given the contributions that our methods and experiments already offer. The Reviewer similarly notes that “this does not negate the innovativeness of this work”. We have now clarified in our Discussion (lines 516-519) and in a new Appendix (D) that the focus of our work is the principle of the method, by highlighting the cost of its current implementation as a key target for improvement with future work, and by offering a number of suggestions and a new analysis exploring how this cost may be reduced.
>
> We also note that the Reviewer gave a 2 for presentation, but did not mention any specific weaknesses related to this. We note that the other Reviewers noted some issues with presentation and clarity, and that we have now addressed those issues. So, we hope that these changes (partially or completely) address the presentation issues the Reviewer had in mind. However, if there are remaining issues of presentation or clarity, we would greatly appreciate it if the Reviewer could offer specific comments regarding those, so that we can address them directly and change things accordingly.

---

> > ### Comment · Reviewer_5yCK · 2024-12-01
> >
> > Thank you for your detailed response. I decide to maintain my score.

---

### Official Review · Reviewer_rCUA · 2024-11-04

**Soundness:** 2
**Presentation:** 2
**Contribution:** 3
**Rating:** 5
**Confidence:** 3

**Summary:**

This paper proposes the Forking Tokens Hypothesis, that there exist some tokens the prefix such that changing them will lead to dramatic differences in the suffix. They use this hypothesis to study uncertainty estimation in the model's output sequences. Technical-wise, they use Bayesian change point detection and survival analysis to identify the location and number of forking tokens.

**Strengths:**

1. The paper formulate a interesting and (I think) novel hypothesis about forking tokens, that there are a few sparse but critical tokens that will determine the trajectory of the generation, and uncertainty estimation should depend on these critical tokens.
2. The estimation method (finding the critical forking token) seems statistically motivated and sound.

**Weaknesses:**

1. I think the method section is written in a very unclear way. For example, the Bayesian formulation in line 258 can be better described. The Gibbs sampling step is also very unclear. There seem to be lots of details missing. A related question: why use linear regression for the CPD? The math in the survival analysis part makes sense, but still lacks all the execution details: what is d? and what are the pros and cons of these two approaches?
2. I agree that the forking theory is interesting, but I think the motivation of the paper is to use it for better uncertainty estimation. but no experiment directly ties these two parts together. It would be great to add an experiment about using the forking tokens to recalibrate the model, and yield higher calibration score. Otherwise, I don't quite buy the connection between forking theory and uncertainty estimation.

**Questions:**

see the weakness section.

---

> ### Author Response · Authors · 2024-11-22
>
> We are glad the Reviewer found our forking tokens hypothesis to be “interesting” and “novel”, and further that they found our methods to be “statistically motivated and sound”. We also thank the Reviewer for their many useful comments, which we respond to below.
>
>
> > 1. I think the method section is written in a very unclear way. For example, the Bayesian formulation in line 258 can be better described. The Gibbs sampling step is also very unclear.
>
> Thank you for pointing out this lack of clarity. We have edited the writing in Section 2 to be more clear, in particular describing the change point detection model (Section 2.3) more succinctly. We also added an Appendix (B) which gives a detailed description of our change point detection model, including the specific implementation [1] which gives more detail on the Gibbs sampling process.
>
>
> > There seem to be lots of details missing.
>
> We now provide additional details of our CPD model in Appendix B. We have also added a new Appendix G to provide details of our LLM sampling pipeline (described in Section 2.1), which includes all prompts and functions used for generating text completions and extracting outcome representations. We believe this added information now provides readers with the details necessary to better understand our methods.
>
>
> > A related question: why use linear regression for the CPD?
>
> We have clarified our particular model choices in App. B. To put it briefly here: linear models match our qualitative observation that there are stable regimes of uncertainty, as well as gradual drift. Linear models also align with our hypothesis that there are sharp changes, without introducing unnecessary complexity. A more complex model, e.g. fitting exponent parameters to t for each segment, might also make it more difficult to interpret abrupt changes in the case where segments have different exponents.
>
> We also note that our work, to our knowledge, is novel in examining and modeling uncertainty dynamics during text generation in this way, and so we have no prior work to compare to when choosing the appropriate statistical model. The closest related work we know of is [2], which analyzes in-weights learning dynamics using HMMs. We see this being an exciting direction for future research, to better understand which statistical models are most appropriate for modeling text generation dynamics.
>
>
>
>
>
> > The math in the survival analysis part makes sense, but still lacks all the execution details: what is d?
>
>
> Thank you for pointing this out. We have modified Section 2.4 to specify how survival analysis can identify forking phenomena, which are different from those identified by our change point detection model. d is an arbitrary vector distance metric, and we have added text to both sections which specifies that we use L2 distance for d in our experiments. We also added an Appendix (F), to clarify the execution detail of our hyper-parameter choice of $\epsilon = .6$ with survival analysis (Figure 7), and our threshold for $p(\tau = t | y)$ when aggregating across examples (Figure 6). This appendix shows how both analysis results change with various thresholds.
>
>
>
>
> >  and what are the pros and cons of these two approaches?
>
>
> We appreciate this suggestion, and have made two key changes to clarify this point. First, we updated the text in Section 2.4 (Lines 296 - 299) to clarify why analyzing o_{t, w} may show a different kind of “forking” than we observe in o_t. Second, we added an analysis in Appendix C.3 that compares the two analysis methods directly. We find that the number of change points predicted by our change point model is not correlated with the final survival rate estimated by our second model. This provides empirical support for our claim that our two methods measure different kinds of forking tokens.
>
>
> *(Continued)*

---

> ### Author Response · Authors · 2024-11-22
>
> *(continued)*
>
> > 2. I agree that the forking theory is interesting, but I think the motivation of the paper is to use it for better uncertainty estimation. but no experiment directly ties these two parts together. It would be great to add an experiment about using the forking tokens to recalibrate the model, and yield higher calibration score. Otherwise, I don't quite buy the connection between forking theory and uncertainty estimation.
>
> We are glad the Review finds our theory interesting. We wish to clarify that our methods are intended for estimating uncertainty in text generation. Calibration, though closely related, is not an objective of this work. Forking tokens and uncertainty dynamics provide new perspectives on uncertainty in text generation, and for this reason our methods cannot be directly compared (“apples-to-apples”) with prior, static methods.  In our experiments, we find many cases of complex uncertainty dynamics that are completely invisible to “static” point estimates of uncertainty such as those used in calibration.
>
> To illustrate these points, we have added Appendix C.1 that compares our outcome distributions $o_t$ to static uncertainty estimates as are used in prior work, such as calibration. We use 3 baselines for static uncertainty estimates: first, we take the log probability of the final answer token in the base path; second, we prompt the model to report a % confidence for the answer in the base path; third, we resample a batch of model responses and compute a simple histogram over outcomes. In Appendix C.1, we show that (a) these baselines may conflict with each other when there are non-trivial uncertainty dynamics, and (b) our methods for analyzing token-level uncertainty can provide insight into why such conflicts might occur.
>
>
>
>
>
> [1] Zhao et al. (2019). Detecting change-point, trend, and seasonality in satellite time series data to track abrupt changes and nonlinear dynamics: A Bayesian ensemble algorithm.
>
> [2] Hu et al. (2023). Latent state models of training dynamics. arXiv:2308.09543.
>
>
>
>
>
>
> **Rebuttal Summary**
>
>
> We appreciate the Reviewer’s comments that our scientific hypothesis is “interesting” and “novel”, and our methods are “statistically motivated and sound”.
>
> They list two main criticisms: first, they find our method section (Sec. 2) to be lacking in  details, and difficult to understand, particularly the sections which describe our statistical models (Sec. 2.3, 2.4). Second, they are unsure of how to relate our hypothesis, as well as our methods and results, with prior work on uncertainty estimation.
>
> We made a number of minor changes to address these weaknesses. We have edited our writing in Sections 2.3 and 2.4 to more clearly describe our analysis methods. To improve replicability, we have specified further details of our change point model in Appendix B, as well as details of our sampling pipeline in Appendix G. We added Appendix F which shows additional analyses that were useful in our selecting hyper-parameters in our models. Lastly, we added Appendix C which compares our two analysis methods to prior uncertainty estimation baselines, as well as to each other.

---

> ### Author Response · Authors · 2024-12-02
>
> Reviewer rCUA,
>
> Thank you for your time and effort in reviewing our paper. We have revised our paper and provided a detailed rebuttal addressing your specific comments and indicating specific parts of the paper with relevant updates.
>
> Please let us know if your questions and concerns have been sufficiently addressed. We would be grateful if you considered raising your score if so, or otherwise provided a response to help us understand which points have or have not been addressed.
>
> We value your feedback, and we appreciate your time spent reviewing our paper.

---

### Official Review · Reviewer_5J1y · 2024-11-04

**Soundness:** 4
**Presentation:** 2
**Contribution:** 4
**Rating:** 8
**Confidence:** 3

**Summary:**

This paper poses an intriguing question: are there time steps $t$ that, if altered, would significantly impact the sequence completion $x_{>t}$? The authors find that such points, referred to as "forking tokens" in their work, do indeed exist. Additionally, they propose a method to automatically detect these forking tokens using a change point detection (CPD) model. More specifically, they derive a time series $y_t = \text{Dist}(o_0, o_t)$, where $o_0$ and $o_t$ represent the expected answer distributions when tokens are changed at time steps 0 and $t$, respectively. They then train two models, $p(\tau = t | y)$ and $p(m \geq 1 | y)$, to predict the likelihood of a forking token occurring at time $t$ and the number of forking tokens, respectively. The authors demonstrate several interesting cases of their method across extensive datasets.

**Strengths:**

1. The research question in this work is compelling because forking tokens are important for understanding model behaviors and steering model generation.
2. The approach of detecting forking tokens with CPD models is also innovative.

**Weaknesses:**

The core contribution of this paper is already intriguing, so the following weakness is likely minor.

The method section is somewhat difficult to follow. In Section 2.2, I struggled due to insufficient explanation of the connection between the definition of $o_t$ and the subsequent detection method at the beginning of Sec. 2.2. While the high-level concept in lines 247–259 is more understandable, some details remain unclear. For instance, defining $\tau_{i-1}$ and $\tau_i$ as the start and end of segment $i$ made it difficult to interpret $\tau$ in $p(\tau = t | y)$. It may affect the reproduce of their method.

**Questions:**

It is also interesting to note that most forking tokens detected are entities. However, certain entities that may be important to humans, such as "June 19, 1972" in Figure 4 (lines 276–277), were not detected by the algorithm. Is there an explanation for this?

---

> ### Author Response · Authors · 2024-11-22
>
> We are encouraged that the Reviewer found our hypothesis to be “intriguing” and “compelling”, our methods to be “innovative”, and our experimental datasets to be “extensive”. We agree that our forking tokens may have “important” implications for understanding and steering text generation with language models.
>
> We also appreciate the Reviewer’s many good points, which we address below.
>
> > They then train two models,  and , to predict the likelihood of a forking token occurring at time  and the number of forking tokens, respectively.
>
> This is correct, we use two kinds of models for analyzing uncertainty dynamics. However, for change point detection we only use a single model, which jointly infers the time and number of forking tokens for each sequence. We have updated our methods section (2.3) and added an appendix for the change point model (App. B) in order to clarify this detail.
>
>
>
> > The method section is somewhat difficult to follow. In Section 2.2, I struggled due to insufficient explanation of the connection between the definition of $o_t$ and the subsequent detection method at the beginning of Sec. 2.2. While the high-level concept in lines 247–259 is more understandable, some details remain unclear. For instance, defining  and  as the start and end of segment  made it difficult to interpret  in .
>
> We thank the Reviewer for pointing out this lack of clarity, and we have updated our methods section to and made it easier to follow. At the end of Section 2.2 we have improved the transition between defining $o_t$ and our subsequent sections on detecting forking tokens. We have updated Section 2.3 to more clearly explain high-level details of the model, and added App. B, which gives a deeper dive into the specifics of the change point model.
>
>
> > It may affect the reproduce of their method.
>
> We share the concern with ensuring that our research is reproducible. For better reproducibility, we have added an appendix (App. B) that goes into further detail regarding the change point model described in Sec. 2.3, including the particular implementation we used. We also added  an appendix (App. G) that provides the specific prompts and outcome parsing functions we used in the sampling pipeline described in Sec. 2.1. We also note that we are committed to releasing, with the camera-ready version of our work, all code used for each step of our methods and all data collected in our experiments, such that anyone interested in reproducing our work can do so easily.
>
>
>
> > It is also interesting to note that most forking tokens detected are entities.
>
> This is an interesting observation, that there may be patterns to which particular tokens are forking tokens. In some cases such as HotpotQA-8076 (Fig. 4) and MMLU-12 (Fig. 5, Top), we find forking tokens at points which are in some sense “expected” – the first time the final answer is explicitly mentioned during the chain of thought. However, in other cases such as GSM8k-59 (Fig. 5, Bottom), we find tokens at unexpected places such as punctuation marks. We have updated the example in Fig. 5 (Bottom) to better demonstrate this point, along the description of these results in the text (Section 4, lines 417-422).
>
> We also hope to add, for the camera-ready version of this work, a preliminary analysis comparing the categories of tokens identified, for example comparing how many forking tokens are ‘content’ words, such as this date, compared to stop words such as “is” and “the” and punctuation marks.
>
>
> > However, certain entities that may be important to humans, such as "June 19, 1972" in Figure 4 (lines 276–277), were not detected by the algorithm. Is there an explanation for this?
>
> The Reviewer raises an interesting question: if humans were given a similar text generation task, which words would people “fork” at, and why? Moreover, if given a single “path”, which tokens would people predict to be forking tokens, and is this correlated with the tokens we find? We consider this point in our discussion (Section 5, lines 529 - 538), which has been edited to improve readability. We also hope to empirically explore these questions in future work.
>
> For the specific date token you mention, we note that this date is directly copied from the prompt, unlike the string “Mia Sara”. If the LM forked at this date token, it would suggest that the model is unreliable in copying information. This would be surprising in light of research demonstrating specific information copying mechanisms in LMs [1]. It is true that if we manually intervened on this token and changed it, subsequent text might change significantly. However, our method instead measures whether such forking tokens are likely to be sampled by the model itself during autoregressive text generation. We have added a sentence in Section 2 (lines 107, 137, 138) which clarifies this point.
>
>
>
> [1] Olsson et al. (2022). In-context learning and induction heads.
>
> *(continued)*

---

> ### Author Response · Authors · 2024-11-22
>
> *(continued)*
>
> **Rebuttal Summary**
>
>
> We are glad the Reviewer found our hypothesis to be “intriguing”, “important”, and “compelling”, our methods to be “innovative”, and our experiments to be “extensive”.
>
> The main weakness that the Reviewer mentions is that parts of our Methods section (Sec. 2) are unclear, and they indicate a few specific examples of this. They also raise a concern about the replicability of our work, in part due to this lack of clarity.
>
> To address these points, we have edited the writing in Sections 2.2, 2.3, and 2.4 to clarify key points and the connections between different parts of our methods. In order to improve the replicability of our methods, we have also added a new Appendix (B) which thoroughly describes our model details, as well as an Appendix (G) which gives specific implementation details for our LLM sampling pipeline.

---

> > ### Comment · Reviewer_5J1y · 2024-11-27
> > **Response to Authors**
> >
> > Many thanks for the detailed response and paper revision. I don't have additional questions.

---

### Author Response · Authors · 2024-11-22
**Note to Reviewers and Area Chair**

**Note to Reviewers and Area Chair**

We are glad the reviewers have judged our work to be “compelling”, “intriguing”, and “very interesting”. We also appreciate their comments that our scientific hypothesis is “novel” and “important”, that our methods for testing this are “innovative” and “sound”, and that our experiments are “extensive” and “comprehensive”.

In addition to their positive remarks, the reviewers raised many useful comments and questions. Reviewers 5J1y and rCUA pointed out difficulties in understanding our methods section (Sec. 2), particularly our change point detection model (Sec. 2.3), and lacked key details for replicability. Reviewer rCUA found the connection between forking tokens and uncertainty estimation unclear, in particular how our method compares to prior work. Reviewer 5yCK found the computational cost of our method to be a potential obstacle to its scalability in future work.

Based on these comments, we have made the following changes, which we believe have significantly improved our paper.

Readability and clarifications: We have made several changes to our paper to improve readability and replicability. The methods section (particularly 2.3 and 2.4) has been improved to more effectively communicate our analysis approach, and we have added appendices which thoroughly describe our change point detection model (App. B) along with specific prompt templates used and other details (App. G). Writing in parts of the Results (Sec. 4), Discussion (Sec. 5) and Introduction (Sec. 1) has also been improved.

Further analysis and experiments: we have added appendices with additional analyses. By comparing our analysis results to prior “static” uncertainty estimation methods (App. C), we show how our analysis is more informative, and can even help explain discrepancies between these methods. We also added the results for our experiments which motivated our choice of hyper-parameters (App. F).

Addressing computational costs: Finally, we have added an appendix (App. D) which includes suggestions for how future work might perform our analysis at a lower cost, as well as an experiment which evaluates how accurately it would perform with fewer samples.

We thank the reviewers for their thoughtful feedback, which has greatly helped us to improve the overall quality of our paper.

---

### Meta-Review · Area_Chair_QAFL · 2024-12-20

**Metareview:**

The paper presents an intriguing phenomenon related to text generation: certain tokens have significant impact on the rest of the sequences.

Reviewers generally agree that the paper is interesting. I believe the work can be followed by: 1) analysis of the cause (is it a property of the token/prefix, or is it a property of the neural network dynamics?), and 2) discussion on the implication of the finding (can we make use of the phenomenon? anything that we should be cautious about?). They may be addressed in future work.

**Additional Comments On Reviewer Discussion:**

Reviewers generally agree that the finding is interesting. Reviewers have different confidence on the presentation quality of the paper, which is relatively minor.

---

### Decision · Program_Chairs · 2025-01-22

Accept (Poster)